# Cortical Up states induce the selective weakening of subthreshold synaptic inputs

Julian Bartram[1,2], Martin C. Kahn[1], Simon Tuohy[1,3], Ole Paulsen [2,4], Tony Wilson[3] & Edward O. Mann [1,2]

Slow-wave sleep is thought to be important for retuning cortical synapses, but the cellular mechanisms remain unresolved. During slow-wave activity, cortical neurons display synchronized transitions between depolarized Up states and hyperpolarized Down states. Here, using recordings from LIII pyramidal neurons from acute slices of mouse medial entorhinal cortex, we find that subthreshold inputs arriving during the Up state undergo synaptic weakening. This does not reflect a process of global synaptic downscaling, as it is dependent on presynaptic spiking, with network state encoded in the synaptically evoked spine $Ca^{2+}$ responses. Our data indicate that the induction of synaptic weakening is under postsynaptic control, as it can be prevented by correlated postsynaptic spiking activity, and depends on postsynaptic NMDA receptors and GSK3β activity. This provides a mechanism by which slow-wave activity might bias synapses towards weakening, while preserving the synaptic connections within active neuronal assemblies.

[1] Department of Physiology, Anatomy and Genetics, University of Oxford, Oxford OX1 3PT, UK. [2] Oxford Ion Channel Initiative, University of Oxford, Oxford OX1 3PT, UK. [3] Department of Engineering Science, University of Oxford, Parks Road, Oxford OX1 3PJ, UK. [4] Department of Physiology, Development and Neuroscience, University of Cambridge, Physiological Laboratory, Cambridge CB2 3EG, UK. Julian Bartram and Martin C. Kahn contributed equally to this work. Correspondence and requests for materials should be addressed to E.O.M. (email: ed.mann@dpag.ox.ac.uk)

During non-rapid eye movement sleep (NREM), local populations of cortical neurons exhibit synchronized membrane potential oscillations between depolarized 'Up states' and hyperpolarized 'Down states'[1]. Up states are intrinsically generated in the cortex via recurrent excitation[2–4], with network activity stabilized through balanced increases in inhibition[4–6]. These patterns of synchronized activity are reflected as slow waves in macroscopic extracellular field potentials, and can propagate across the cerebral cortex as travelling waves[4, 7, 8], recruiting the majority of neurons in neocortex and paleocortex[9]. The power and incidence of slow-wave activity (0.5–4 Hz in the electroencephalogram) change across development[10, 11], and are under global and local homeostatic control, suggesting that slow waves may serve an important role in cortical circuit function.

Slow-wave activity increases during infancy and preadolescence, and subsequently decreases over the course of natural maturation[10–13]. This inverted U-shaped profile has correlates with the density and strength of synaptic connectivity in cortical circuits, paralleling consecutive periods of synaptogenesis, pruning, and synaptic reorganization. It may be that slow waves not only reflect the functional architecture of cortical circuits, but also provide a self-generated substrate for offline retuning following experience. Indeed, even after cortical maturation is complete, there remains a global homeostatic modulation of slow-wave activity, which is high after sustained wakefulness, and dissipates across NREM sleep episodes[14, 15]. There are also region-specific increases in cortical slow-wave activity following experience and learning[16, 17], and both natural and induced increases in slow-wave oscillations can improve subsequent performance in learned tasks[16, 18].

While there is strong evidence that slow-wave activity promotes memory consolidation, particularly for declarative memories involving the temporal lobe system[19], the underlying cellular and circuit mechanisms have yet to be resolved. One prominent model suggests that slow-wave activity drives the reactivation of memory traces across hippocampal-parahippocampal-neocortical networks, and thus supports the redistribution and transformation of new memories for long-term storage[19–21]. An alternative hypothesis is that slow-wave activity induces a global downscaling of synaptic weights across cortical

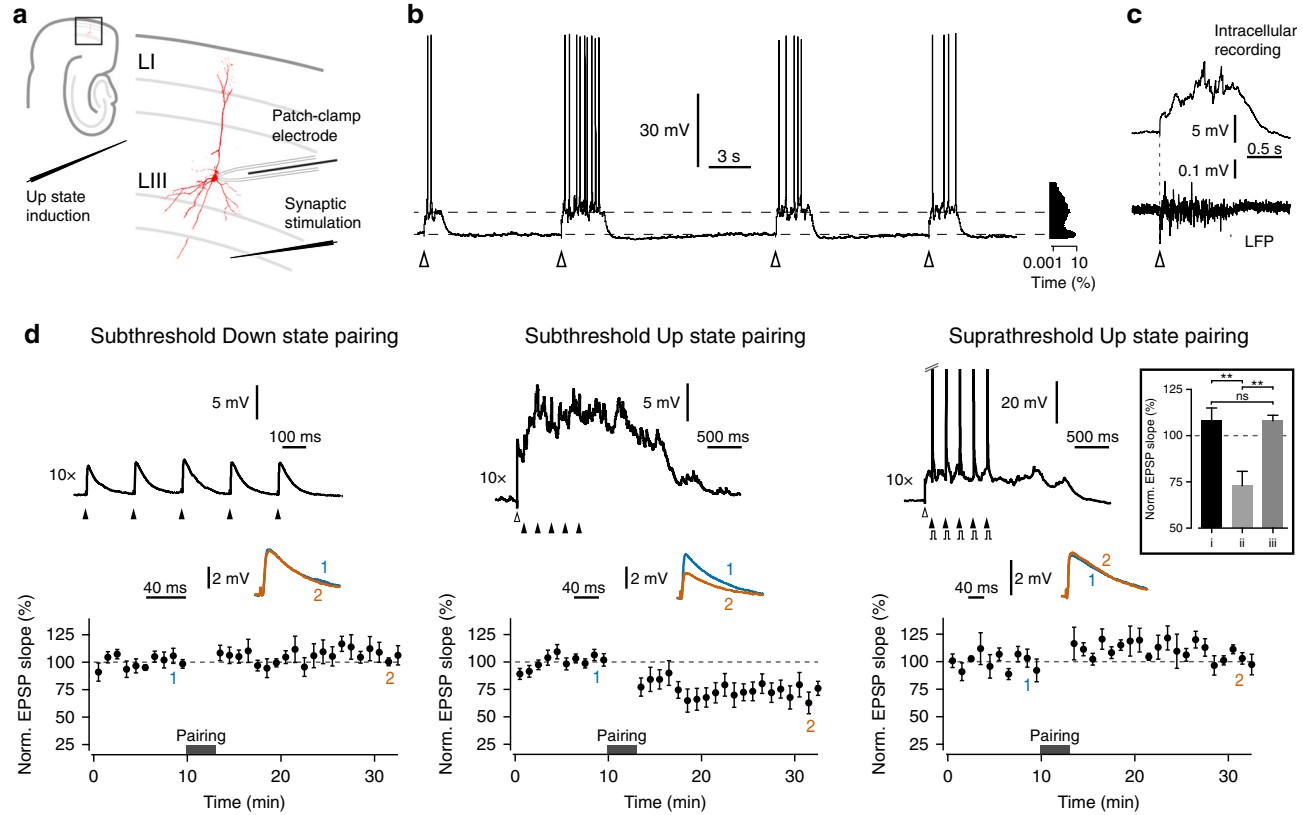

**Fig. 1** Pairing subthreshold inputs with Up states causes synaptic weakening. **a** Schematic of the experimental set-up, with whole-cell patch-clamp recordings obtained from a LIII principal neuron, and extracellular stimulation electrodes placed in LIII and the border of the deep layers, to evoke local Up states and single EPSPs, respectively. **b** Current-clamp recording from a layer III principal cell shows reliable induction of Up states following electrical stimulation in LIII (*white arrowheads*), resulting in a typical bimodal distribution of the membrane potential (*right histogram*; percentage of time on a logarithmic scale). **c** Simultaneous whole-cell current-clamp (*upper trace*) and LFP (*lower trace*) recordings from layer III showing an induced Up state. **d** Single EPSPs were evoked every 15 s during Down state periods in order to measure synaptic strength during baseline, and following three pairing protocols with 5 Hz stimulus trains: Down state pairing (total of 50 individual EPSP pairings; *n* = 6; *left panel*), Up state pairing (total of 50 individual EPSP pairings; *n* = 9; *middle panel*), and Up state pre-post spike pairing (total of 50–100 individual EPSP pairings; *n* = 6; *right panel*). *Top*: example recordings during pairing protocols, showing the timings of the synaptic stimuli (*black arrowheads*), Up state induction (*white arrowheads*), and the brief somatic current injections used to evoke spiking following each synaptic stimulation in the Up state pre-post pairing protocol (*grey steps*). Spontaneous spikes did occur during Up states, and the example traces with no background spiking are chosen to emphasize that responses to synaptic stimulation were predominantly subthreshold during pairing, which is further detailed in Supplementary Fig. 1. *Middle*: representative EPSP traces from the baseline period (1; *blue*) and the end of recording (2; *orange*) (mean of five traces). *Bottom*: pooled data of normalized single EPSP slope across experiments for each pairing protocol (1 min bins). *Inset*: summary statistics, showing that only the subthreshold Up state pairing induced synaptic weakening. **P < 0.01, One-Way ANOVA followed by Tukey's multiple comparisons test. *Error bars* show the SEM

networks, which is necessary to compensate for the net build-up of synaptic weight as a result of experience and learning during the preceding waking period[22]. It has been suggested that this synaptic homeostasis could subsequently promote memory recall by increasing the signal-to-noise ratio of synaptic communication.

A role for slow-wave activity in neuronal reactivation and memory consolidation is not mutually exclusive with a more global process of synaptic homeostasis. Indeed, these hypotheses have been difficult to tease apart, with numerous studies investigating the effect of sleep on synaptic plasticity producing mixed results, with evidence for synaptic depression, but also indications of synaptic potentiation[23–25]. In order to resolve this, it will be necessary to elucidate the specific rules of synaptic plasticity operating during cortical slow-wave activity. Here, we studied these mechanisms of plasticity in acute-slice preparations in vitro, which preserve Up/Down states[4, 26, 27], while enabling exquisite control over state transitions and synaptic activation[5, 27]. In rodents, slow-wave activity emerges in the second postnatal week[10], and already begins to decline by the end of the third postnatal week[13]. We studied synaptic plasticity during this preadolescent period of high slow-wave activity (P14-21), focusing on the mouse medial entorhinal cortex (mEC), which is a key interface between the hippocampus and neocortex. We demonstrate that the depolarized Up state phase can induce synaptic weakening of subthreshold inputs, which is counteracted by correlated pre- and postsynaptic spiking activity.

## Results

### Synaptic weakening and maintenance during mEC Up states.
We performed whole-cell current-clamp recordings from mEC layer III principal cells and controlled the precise timing of network Up states and synaptic activation by local electrical stimulation (Fig. 1a). Spontaneous Up states were sparse or absent, while controlled Up state induction was reliable (Fig. 1b). The coinciding membrane depolarization, increase in membrane potential variability and activity in the simultaneously recorded local field potential from layer III confirmed that the observed Up states are in fact a cellular manifestation of a network-driven event (Fig. 1c).

Since membrane depolarization and background synaptic activity are well-known factors influencing synaptic plasticity, we sought to determine the effect of pairing synaptic inputs with network Up states (Fig. 1d). To monitor synaptic efficacy, excitatory postsynaptic potentials (EPSPs) were evoked every 15 s during Down states. As evoked waveforms often contained disynaptic inhibitory components, the slope of the EPSP was used as a measure of excitatory synaptic strength. Following the recording of a baseline period, we applied 5 Hz stimulus trains, to approximate the Up state spike rates of mEC layer III principal cells in vivo[28]. The 5 Hz trains were applied in three different plasticity protocols, which had significantly different effects on the EPSP slope measured 10–20 min post-pairing ($F_{(2, 18)} = 9.63$, $P < 0.01$, $\eta^2 = 0.52$; One-way analysis of variance (ANOVA)): (i) In control experiments, 5 Hz stimulus trains were applied during the Down state, for a total of 50 pairings. Following this synaptic activation, the EPSP slope changed only marginally to $108.4 \pm 6.5\%$ of the baseline value ($n = 6$). (ii) When 5 Hz synaptic stimulation trains were paired with Up states, the slope was significantly reduced to $72.9 \pm 7.9\%$ ($n = 9$; $P < 0.01$ cf Down state pairing, One-way ANOVA followed by Tukey's post hoc test; Cohen's $d = 1.95$). Despite the Up states being associated with membrane depolarization ($9.2 \pm 1.3$ mV, $n = 9$; measured prior to train stimulation) and spiking activity (mean: $1.3 \pm 0.6$ Hz, range: 0.0–4.4 Hz, $n = 9$), the pairings themselves typically evoked subthreshold responses, consistent with the in vivo observation

that sensory-evoked responses are less effective in driving spiking during Up states compared to Down states[29]. Indeed, the pairing during the Up state induced a post-stimulus suppression of spontaneous spiking for up to 45 ms (Supplementary Fig. 1), most likely due to synchronized disynaptic inhibition. Synaptic weakening following pairing did not strongly correlate with either the background Up state spike rate ($R^2 = 0.08$, $P = 0.46$, $n = 9$) or the strength of the evoked response (measured as baseline EPSP slope; $R^2 = 0.13$, $P = 0.35$, $n = 9$; Pearson linear correlation test). (iii) To determine whether suprathreshold spike pairings could influence plasticity during Up states, 5 Hz trains were again paired with evoked Up states (pre-stimulus amplitude: $9.0 \pm 1.6$ mV, $n = 6$), in combination with brief depolarizing current steps to trigger typically one, occasionally two, postsynaptic spikes within the 15 ms period following synaptic stimulation. Correlated pre- and postsynaptic spiking was found to prevent Up state-induced synaptic weakening, with the change of the EPSP slope of $108.0 \pm 3.0\%$ comparable to that observed after Down state pairings ($n = 6$; n.s. cf Down state pairing, Cohen's $d = 0.02$; $P < 0.01$ cf subthreshold Up state pairing, Cohen's $d = 1.92$; One-way ANOVA followed by Tukey's post hoc test).

The synaptic weakening induced by subthreshold Up state pairings was not specific to 5 Hz stimulation trains, and could also be induced by pairing at 2.5 Hz ($72.2 \pm 5.2\%$ of baseline EPSP slope, $n = 6$, $t_{(5)} = 5.35$, $P < 0.01$, one-sample $t$-test) and 7.5 Hz ($72.0 \pm 6.0\%$ of baseline EPSP slope, $n = 7$, $t_{(6)} = 4.63$, $P < 0.01$, one-sample $t$-test). This suggests that synaptic weakening can occur across a range of stimulation frequencies that approximates the range of Up state spike rates observed in vivo[28].

High frequency bursting following synaptic inputs can be particularly effective in inducing synaptic plasticity[30, 31], and might thus enable synaptic potentiation during Up state periods. In 4 of the subthreshold pairing experiments, input-burst pairings during Up state periods were applied following the stabilization of synaptic weakening, and were found to reliably reverse depression, with a trend towards potentiation ($85.3 \pm 4.9\%$ following subthreshold pairing compared to $124.0 \pm 9.3\%$ following burst pairing, measured relative to the initial baseline EPSP slope; $t_{(3)} = 4.00$, $P < 0.05$, paired $t$-test; Supplementary Fig. 2). We next wanted to test whether input-burst pairings during Up state periods could induce potentiation at naïve synapses, and whether this in turn could be reversed by subsequent subthreshold pairing. Pairing 5 Hz stimulus trains with spike bursts during the Up state, for a total of 100 pairings, led to an increase in synaptic strength that did not stabilize within the 20 min post-pairing period (Fig. 2). We therefore performed two sets of interleaved experiments—in the first group synaptic responses were measured for >40 min after input-burst pairing ($n = 7$), and in the second group an additional subthreshold Up state pairing was applied 20 min after input-burst pairing ($n = 6$). These experiments revealed a significant decrease in normalized EPSP slope measured at 10–20 min and 30–40 min post input-burst pairing ($F_{(1,11)} = 8.57$, $P < 0.05$; $\eta^2 = 0.48$), but no significant differences between the pairing paradigms ($F_{(1,11)} = 0.01$, $P = 0.96$; $\eta^2 = 0.001$), or interaction between time and pairing paradigm ($F_{(1,11)} = 0.02$, $P = 0.90$; $\eta^2 = 0.001$; mixed ANOVA). Across the groups, the normalized EPSP slope was significantly potentiated at 10–20 min ($158.5 \pm 10.5\%$, $n = 13$, $t_{(12)} = 5.47$, $P < 0.001$) and 30–40 min post input-burst pairing ($133.5 \pm 12.8\%$, $n = 13$, $t_{(12)} = 2.62$, $P < 0.05$; one sample $t$-tests, with Bonferroni correction for multiple comparisons; Fig. 2).

These results suggest that inputs that consistently evoke subthreshold responses during Up states undergo synaptic weakening, while suprathreshold inputs are preserved, protected, and sometimes strengthened. However, with our extracellular stimulation protocol, we were not able to determine the source of

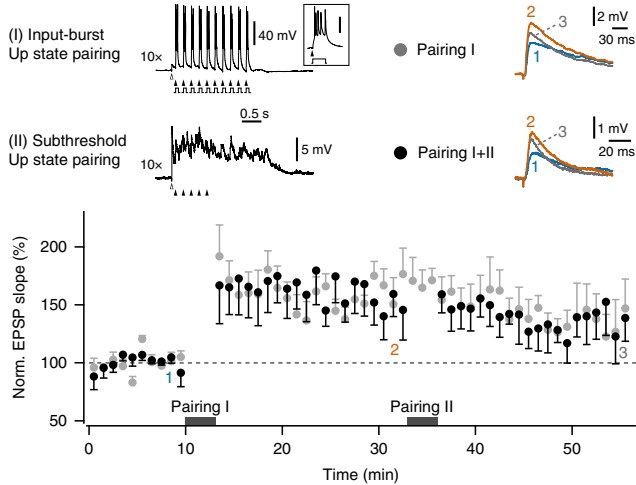

**Fig. 2** Pairing subthreshold inputs with Up states does not reverse synaptic potentiation. Pairing synaptic input with spike bursts during Up states induced synaptic potentiation (Pairing I: stimulus trains at 5 Hz, 100 pairings in total; each extracellular stimulation was followed by a depolarizing current step evoking >3 postsynaptic spikes). In one group of experiments, synaptic responses were measured for >40 min after input-burst pairing, and in the second group an additional subthreshold Up state pairing (Pairing II: stimulus trains at 5 Hz, 50 pairings in total; same as applied in Fig. 1d) was applied 20 min after input-burst pairing. *Top, left*: representative recordings during Pairing I (*upper*) and Pairing II (*lower*), showing the timings of Up state induction (*white arrowheads*), synaptic stimulation (*black arrowheads*), and the brief somatic current injections used to evoke spike bursts (*grey steps*). The *inset* shows one magnified input-burst pairing (*scale bar*: 40 mV). *Top, right*: EPSPs from the baseline (1), 20 min after Pairing I (2) and > 40 min after Pairing I (3), for an experiment with Pairing I only (*upper*) and Pairing I + II (*lower*; average of five traces each). *Bottom*: pooled data of normalized single EPSP slope across experiments for each pairing protocol (1 min bins), showing a decrease in synaptic potentiation over time, but no significant effect of pairing paradigm or significant interaction between time and pairing paradigm (results of mixed measures ANOVA reported in text). *Error bars* show the SEM

the stimulated fibers, their spontaneous activity during Up states, or whether the plasticity reflects a global scaling of synaptic inputs. In order to overcome the inherent limitations of extracellular stimulation, and further understand how patterns of pre- and postsynaptic activity control plasticity during network Up states, it was necessary to perform paired-recordings.

**Synaptic weakening by spontaneous Up state spiking**. Extracellular stimulation at the border of the deep layers in mEC is likely to antidromically activate recurrent connections between LIII pyramidal neurons. In order to determine whether Up states could induce weakening at these specific synapses, we performed dual whole-cell patch-clamp recordings from synaptically coupled mEC LIII principal cells (Fig. 3a). Pairs were arbitrarily assigned to one of two groups: in the first group, hyperpolarizing current injection in the presynaptic cell was used to suppress presynaptic spikes during Up states (termed 'presynaptic suppression group', mean spike rate: $0.041 \pm 0.028$ Hz, $n = 5$; Fig. 3b). This manipulation was designed to test whether it might be possible for inactive synapses to undergo weakening during Up states, due to correlated changes in pre- and postsynaptic membrane potential, heterosynaptic interactions and/or a process of global downscaling. In the second group, the presynaptic cell exhibited Up state-associated spiking at varying rates (mean spike rate: $4.1 \pm 0.6$ Hz, $n = 6$; Fig. 3c). Postsynaptic Up state spiking was not

controlled in either group. Unitary postsynaptic EPSPs were evoked throughout the recording every 15 s by presynaptic spikes during Down states, and the response amplitudes were measured. After a baseline period, Up states were triggered once every 15 s for ~15 min. In the presynaptic suppression group (Fig. 3b), the EPSP amplitude changed on average only marginally to $96.5 \pm 3.8\%$ ($n = 5$) of the baseline value. In the group of pairs with spontaneous presynaptic Up state spiking, however, a gradual reduction in synaptic strength started shortly after Up state induction, and the inputs remained weakened until the recording end (Fig. 3c). This synaptic weakening to $71.1 \pm 7.0\%$ of the baseline value ($n = 6$) was significantly different from the presynaptic suppression group ($t_{(9)} = 3.00$, $P < 0.05$, unpaired $t$-test; Fig. 3d). The results confirm that Up state-dependent weakening of synaptic inputs occurs at recurrent connections within mEC layer III, and demonstrate that the induction can be achieved by spontaneous presynaptic spike patterns. The results also demonstrate that Up state activity alone is not sufficient to modulate synaptic weights, and are consistent with this form of plasticity being input-specific.

As we aimed to explore the effects of naturally evolving spike patterns during Up states, the precise number of presynaptic spike pairings was not controlled. In order to explore whether this might explain some of the variations in the degree of synaptic weakening across experiments, the EPSP amplitude after Up state induction (average of the last 10 min period) was plotted against the mean presynaptic Up state spike rate (Fig. 3e). For the presynaptic suppression group, in which presynaptic spiking was low or absent, there appeared to be a small trend towards synaptic weakening in the experiments showing some presynaptic spikes. However, in the group with spontaneous presynaptic spiking, synaptic weakening became less prominent with higher presynaptic spike rates ($R^2 = 0.76$, $n = 6$, $P < 0.05$; Pearson linear correlation test). This at first appeared inconsistent with the results of the experiments with extracellular stimulation, in which synaptic weakening could be induced by stimulation at 2.5, 5, and 7.5 Hz. However, pre- and postsynaptic Up state spike rates were correlated ($R^2 = 0.88$, $n = 6$, $P < 0.01$; Pearson linear correlation test; Fig. 3f), and hence synaptic weakening also became less prominent with higher postsynaptic Up state spike rates ($R^2 = 0.68$, $P < 0.05$, $n = 6$; Pearson linear correlation test). The timing of postsynaptic spikes displayed a relatively flat distribution around each presynaptic spike in the Up state (see inset in Fig. 3f), suggesting that increasing Up state spike rates promote spike pairings, and could thus counteract synaptic weakening. A role for spike timing is also consistent with the previous experiments showing that synaptic weakening was prevented by analogous induced suprathreshold pairings (Fig. 1d).

To explore the locus of expression of synaptic weakening, we performed a coefficient of variation (CV) analysis of the paired recording data. The CV of EPSP amplitude increased from $0.26 \pm 0.04$ (baseline) to $0.33 \pm 0.05$ (following Up state pairing; $n = 6$), and plots of the normalized $CV^{-2}$ and the normalized mean EPSP amplitude showed that most points clustered around the line of unity (Supplementary Fig. 3a). We also applied the same analysis to the EPSP slope measurements recorded during subthreshold Up state pairing with extracellular stimulation (Fig. 1d), and found similar results (CV baseline: $0.26 \pm 0.02$; CV following Up state pairing: $0.34 \pm 0.05$; $n = 9$; Supplementary Fig. 3b). Changes in normalized $CV^{-2}$ are indicative of altered presynaptic release (see also Supplementary Fig. 3c), but can also be produced by postsynaptic mechanisms, including the changes in the number of silent synapses[32]. We therefore performed an additional series of experiments using paired pulse extracellular stimulation (inter-stimulus interval: 50 ms), and found that paired pulse facilitation increased

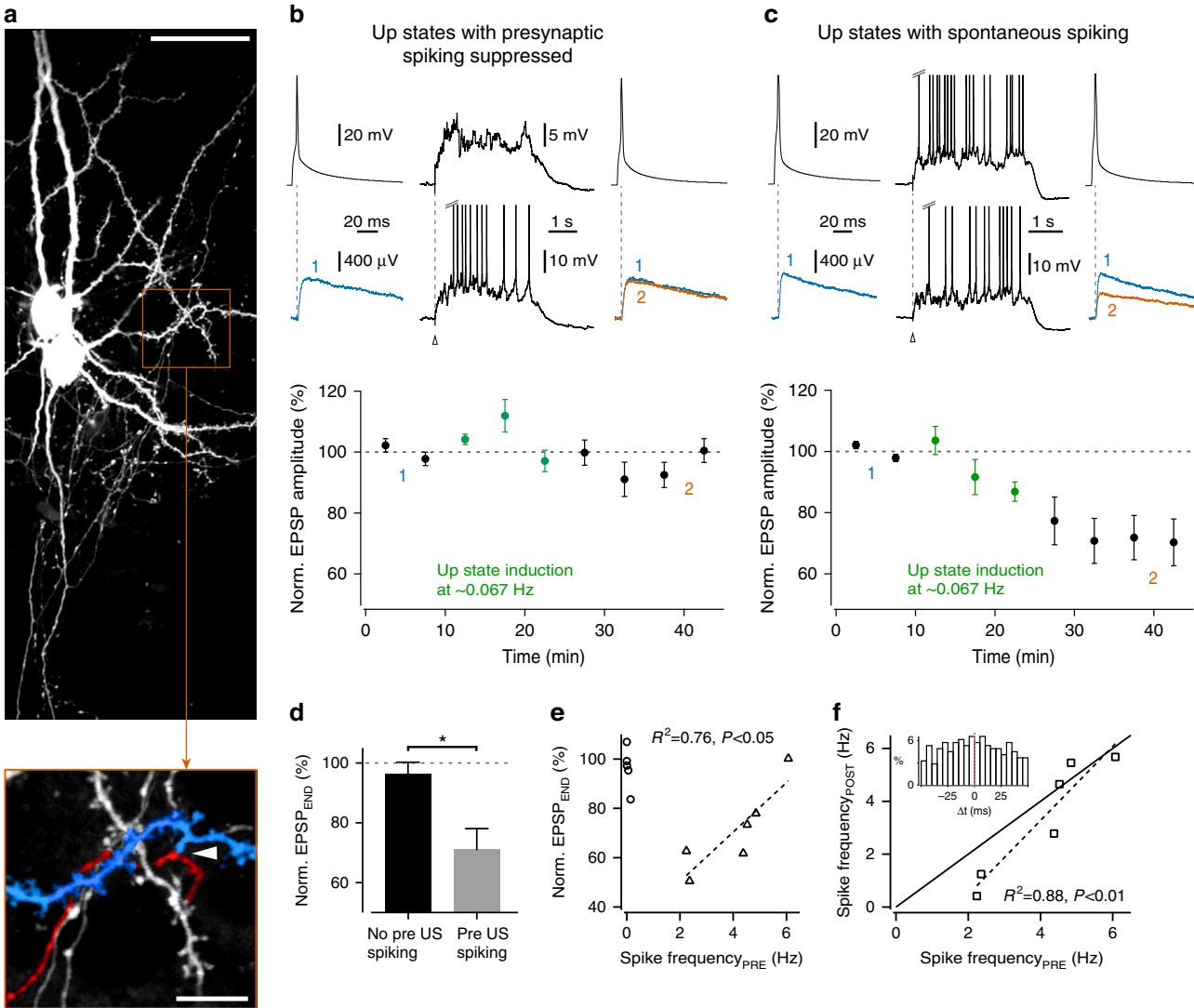

**Fig. 3** Spontaneous spike patterns generated during Up states (US) induce synaptic weakening in a frequency-dependent manner. **a** Two-photon reconstruction of two synaptically coupled mEC LIII principal cells labeled with biocytin. A putative synapse formed between a presynaptic axonal bouton (highlighted in *red*) and a postsynaptic basal dendrite (*blue*) is indicated in the magnified region below (*white arrowhead*). *Scale bars*: 40 μm/10 μm. **b** Presynaptic spikes were triggered by brief (3–5 ms) depolarizing current steps and the postsynaptic EPSP amplitudes were measured. Following the recording of a 10-min baseline, Up states were also induced once per sweep for a period of ~15 min. When necessary, a constant hyperpolarizing current injection was applied to the presynaptic neuron to keep Up state spiking activity low or absent. *Top*: representative traces of evoked presynaptic (*upper*) and postsynaptic (*lower*) activity, during the baseline period (*left*, averages of ten traces), Up state induction (*middle*; timing of Up state induction shown with *white arrowhead*), and at the recording end (*right*, averages of ten traces; baseline EPSP trace included in *blue* for comparison). *Bottom*: time course of changes in normalized EPSP amplitude ($n = 5$; 5 min bins). Period of Up state induction is highlighted in *green*, with numbers indicating the time points used to calculate the average traces. **c** As in **b**, except with presynaptic Up state depolarization sufficient to produce self-generated spike patterns of varying frequencies ($n = 6$). **d** Summary statistics. *$P < 0.05$, unpaired *t*-test. **e** Plot of the relationship between synaptic weakening and presynaptic Up state spike rate (*circles*: control group with subthreshold/sparsely spiking presynaptic Up states; *triangles*: group with presynaptic Up state spiking). For the group with spontaneous presynaptic Up state spiking, the degree of synaptic weakening correlated significantly with presynaptic spike rate (Pearson linear correlation test). **f** For the group with spontaneous presynaptic Up state spiking, there was a significant correlation between postsynaptic and presynaptic Up state spike rates (*dotted line*, Pearson linear correlation test). *Bold line* indicates unity correlation. *Inset*: distribution of postsynaptic Up state spike timing relative to presynaptic activity from recording shown in **c**. Error bars show the SEM

significantly following Up state pairing (measured as ratio of EPSP slopes; $1.35 \pm 0.11$ during baseline compared to $1.69 \pm 0.19$ following pairing; $n = 7$, $t_{(6)} = 2.80$, $P < 0.05$, paired *t*-test; Supplementary Fig. 3d), and that the change in paired pulse ratio correlated significantly with the degree of synaptic weakening ($R^2 = 0.73$, $n = 7$, $P < 0.05$; Pearson linear correlation test; Supplementary Fig. 3d). Together, these findings are consistent with a presynaptic component to Up state-associated synaptic weakening.

**State dependence of synaptically evoked spine Ca$^{2+}$ responses.** Previous studies in quiescent slice preparations of neocortex and hippocampus have found that coincident pre- and postsynaptic activity can induce synaptic depression, with similar indicators of a presynaptic component of expression, and have shown that this plasticity is dependent on increases in postsynaptic Ca$^{2+}$ concentration[33–35]. Depolarization during network Up states might be expected to boost postsynaptic Ca$^{2+}$ signaling, and thus enable state-dependent synaptic weakening. However, predicting

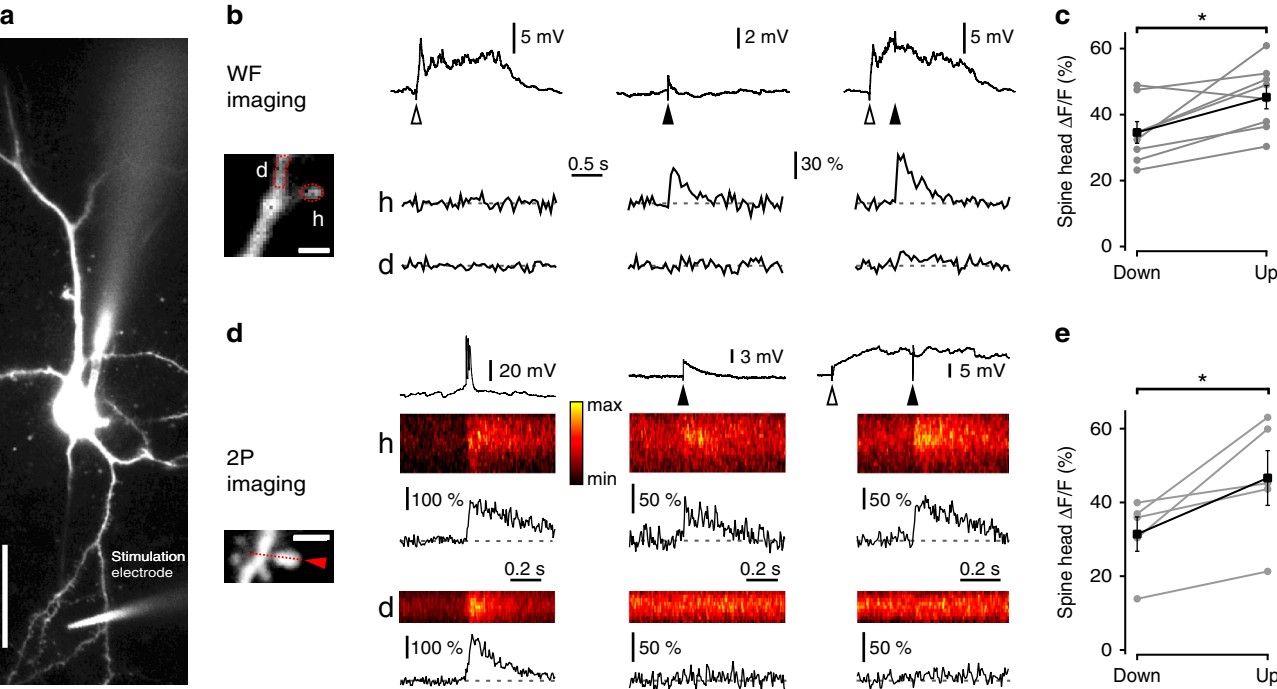

**Fig. 4** Boost of the mean synaptically evoked spine Ca$^{2+}$ response during Up states. **a** Experimental set-up for spine Ca$^{2+}$ imaging, local synaptic activation and Up state induction (stimulation electrode for the latter not in the field of view). *Scale bar*: 40 μm. **b** In wide-field (WF) imaging experiments, the cell was loaded with Alexa 594 (40 μM) and OGB (200 μM), for structural and Ca$^{2+}$ imaging, respectively. The *inset* shows the regions of interest for imaging Ca$^{2+}$ responses in the spine head (h) and neighboring dendritic shaft (d). *Scale bars*: 1.5 μm. Somatic voltage responses (*top traces*), and corresponding relative changes in OGB fluorescence (Δ*F/F*) in the spine head (h; *middle traces*), and adjacent dendritic shaft (d; *bottom traces*) evoked by subthreshold Up states (*left panel*), local synaptic stimulation during Down states (*middle panel*), and local synaptic stimulation during Up states (*right panel*). **c** Comparison of the mean spine Δ*F/F* responses (400 ms window) evoked by local synaptic stimulation in Down states vs. Up states across different neurons in wide-field imaging experiments (*n* = 8). **d** In two-photon (2P) imaging experiments, the cell was loaded with Alexa 594 (40 μM) and Fluo-5F (400 μM), for structural and Ca$^{2+}$ imaging, respectively. The inset shows the line-scan path. Example images of line-scans were filtered with a 3 × 3 Gaussian kernel. *Scale bar*: 2.5 μm. Somatic voltage responses (*top traces*), and line scans and Δ*F/F* for Fluo-5F signal in the spine head (h; *middle traces*), and adjacent dendritic shaft (d; *bottom traces*) evoked by spontaneous Up state spike burst (*left panel*), local synaptic stimulation during Down states (*middle panel*), and local synaptic stimulation during Up states (*right panel*). The line-scan plots are scaled between the minimum (min) and maximum (max), and color-coded according to the *inset* key. **e** Comparison of the mean spine Δ*F/F* responses (400 ms window) evoked by local synaptic stimulation in Down states vs. Up states across different neurons in two-photon imaging experiments (*n* = 5). *P < 0.05, paired *t*-test. *Error bars* show the SEM

the effect of Up states on synaptically evoked Ca$^{2+}$ responses is not straightforward, as they involve the coordinated firing of excitatory and inhibitory neurons, and, for example, co-active spine-targeting interneurons could selectively inhibit local Ca$^{2+}$ signaling[36]. To investigate the influence of Up states on spine Ca$^{2+}$ responses, we combined wide-field spine Ca$^{2+}$ imaging with local electrical activation of synapses (Fig. 4a, b). mEC layer III-principal cells were loaded with the calcium indicator OGB (200 μM) for at least 30 min via a patch-pipette. Another micro-electrode was positioned near a basal dendrite (~5–20 μm distance to imaged spine) for local synaptic activation (Fig. 4a). We targeted basal dendritic locations where we had previously observed LIII-LIII synapses (Fig. 3a). Cells were minimally hyperpolarized to minimize Up state spiking, and prevent contamination of the recorded postsynaptic Ca$^{2+}$ transients by back-propagating action potentials. Evoking subthreshold Up states did not induce discernible Ca$^{2+}$ transients in the target spine (Fig. 4b). In contrast, local synaptic stimulation during Down states resulted in EPSPs and Ca$^{2+}$ transients in the spine head, while the adjacent dendritic shaft showed only a marginal change in fluorescence (Fig. 4b). Synaptic activation of the same input during Up states typically produced a smaller change in the somatic membrane potential, but also reliably evoked Ca$^{2+}$ responses that were mostly restricted to the spine head (Fig. 4b). To quantify the state dependence of spine Ca$^{2+}$ transients, synaptic activation was alternately triggered during Down state and Up state periods, and the mean signals evoked during successful synaptic transmissions were then compared (Fig. 4c). The average Ca$^{2+}$ response evoked during Up states (Δ*F/F*, 45.3 ± 3.5%) was significantly increased compared to the average Down state response (Δ*F/F*, 34.6 ± 3.3%; *n* = 8 spines from 8 different cells, $t_{(7)}$ = 3.16, *P* < 0.05, paired *t*-test). Although we did not observe prominent Ca$^{2+}$ responses in dendritic shafts during Up states, the wide-field imaging configuration is unable to eliminate scatter and out-of-focus interference. We therefore performed the same stimulation paradigm using two-photon spine Ca$^{2+}$ imaging (Fig. 4a, d), in mEC layer III principal cells loaded with the calcium indicator Fluo-5F (400 μM) for at least 30 min via a patch-pipette. In some recordings, spontaneous spikes were recorded, which as expected, elicited Ca$^{2+}$ responses in both the spine head and neighboring dendritic shaft. However, the synaptically evoked Ca$^{2+}$ responses were mostly restricted to the spine head (Fig. 4d). Moreover, the average Ca$^{2+}$ response evoked during Up states (Δ*F/F*, 46.6 ± 7.4%) was again found to be consistently and significantly increased compared to the average Down state response (Δ*F/F*, 31.4 ± 4.6%; *n* = 5 spines from 5 different cells, $t_{(4)}$ = 2.93, *P* < 0.05, paired *t*-test; Fig. 4e). Thus, both the wide-field and two-photon imaging data show that synaptically activated dendritic spines have information about the network state encoded in their mean Ca$^{2+}$ response during the cortical Up states.

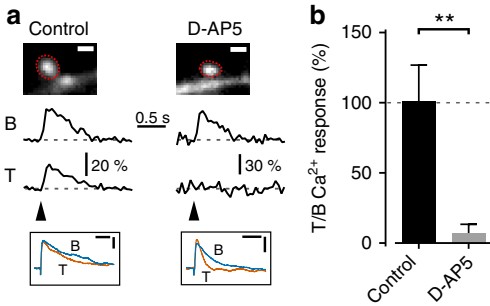

**Fig. 5** Synaptically evoked spine Ca²⁺ transients depend on NMDAR. **a** Wide-field imaging experiments were used to pharmacologically examine the contribution of NMDAR to spine Ca²⁺ transients evoked by local synaptic stimulation, with the cells loaded with Alexa 594 (40 μM) and OGB (200 μM), for structural and Ca²⁺ imaging, respectively. **a** To mitigate the potential effects of run-down over the time course of drug application, synaptically evoked Ca²⁺ transients were imaged during a baseline period (B), and following treatment (T) with either control aCSF (*left*) or bath-applied D-AP5 (40 μM; *right*). *Top*: regions of interest for imaging Ca²⁺ response in the spine head. *Middle*: relative changes in OGB fluorescence (Δ*F*/*F*) in the spine head following local synaptic stimulation during baseline and treatment conditions. *Scale bars*: 1 μm. *Bottom*: corresponding EPSPs evoked by local synaptic stimulation. *Scale bars*: 50 ms/1 mV (control) and 40 ms/2 mV (D-AP5). **b** Pooled data showing the mean spine Δ*F*/*F* responses (400 ms window) in the treatment condition relative to baseline. **P < 0.01, paired *t*-test. *Error bars* show the SEM

Synaptically evoked Ca²⁺ transients in spine heads have previously been shown to depend on activation of *N*-methyl-ᴅ-aspartate receptors (NMDAR)[37–40]. We examined whether NMDAR also contribute to spine Ca²⁺ transients in mEC layer III principal cells using the wide-field imaging set-up (Fig. 5a). Blocking NMDAR with bath application of 40 μM D-AP5 was found to largely eliminate spine Ca²⁺ signals (mean Δ*F*/*F* measured relative to baseline period; control: 101.0 ± 25.8% vs. D-AP5: 7.2 ± 6.3%; *n* = 5 spines from 5 different cells in each group, $t_{(8)} = 3.54$, $P < 0.05$, unpaired *t*-test; Fig. 5b). Under our conditions, this also blocked evoked Up states, which is consistent with a role for NMDAR in Up state initiation in acute slices of mEC[41]. However, this data does suggest that NMDAR drive synaptically evoked spine Ca²⁺ transients, although other Ca²⁺ sources are likely to contribute to these spine transients, and their additional boost during Up states.

**Postsynaptic mechanisms regulating synaptic weakening**. As NMDAR provide a major source for synaptically evoked Ca²⁺ transients in spine heads[37–40], and these transients are boosted in Up states relative to Down states (Figs. 4 and 5), we tested the role of postsynaptic NMDAR in Up state-dependent synaptic weakening using single-cell pharmacology. We repeated the subthreshold Up state pairing protocol (Fig. 1d) with two experimental groups: in the control group the patch-pipette was filled with standard internal solution, while in the second group the NMDAR antagonist MK-801 was additionally included (Fig. 6a). The inhibitor and control internal solutions were given ~30 min for equilibration after breakthrough, before recording baseline responses. Following Up state pairing, synaptic weakening to 68.8 ± 11.7% (*n* = 5) in the control group was significantly reduced to 94.42 ± 3.3% (*n* = 6) in the inhibitor group ($t_{(9)} = 2.30$, $P < 0.05$, unpaired *t*-test). This result supports a role for postsynaptic NMDAR in Up state-induced plasticity, but it is possible that presynaptic NMDAR could have been exposed to MK-801 during the patching procedure, and not recovered from use-dependent block over the approximately

40 min of equilibration and baseline recording. We used two approaches to explore this: (i) Glutamatergic terminals in the entorhinal cortex are thought to express GluN2B subunit-containing NMDAR, which tonically facilitate glutamate release[42]. We first confirmed that washing on the GluN2B subunit-selective antagonist Ro 25-6981 (0.5 μM) reduced the slope of evoked EPSPs in layer III pyramidal neurons (64.3 ± 9.6% of baseline, *n* = 6, $t_{(5)} = 3.72$, $P < 0.05$, one sample *t*-test; Supplementary Fig. 4a). In the presence of 0.5 μM Ro 25-6981, it was still possible to evoke Up states, and the subthreshold Up state pairing protocol (Fig. 1d) was found to induce synaptic weakening to 78.2 ± 4.9% (*n* = 4, $t_{(3)} = 4.48$, $P < 0.05$; one sample *t*-test), which was not significantly different from that obtained during inter-leaved vehicle controls (81.8 ± 4.7%, *n* = 4; $t_{(6)} = 0.54$, $P = 0.69$; unpaired *t*-test; Supplementary Fig. 4b). This suggests that inhibiting presynaptic NMDAR does not prevent Up state-induced synaptic weakening. (ii) As NMDAR are sensitive to voltage-dependent Mg²⁺ block, we also tested whether subthreshold postsynaptic depolarization would be sufficient to induce synaptic weakening in the preadolescent mEC layer III pyramidal neurons. Pairing 5 Hz trains with depolarizing current steps, to mimic the somatic depolarization observed during Up states, induced a significant reduction in synaptic strength to 88.9 ± 3.6% (*n* = 5, $t_{(4)} = 3.10$, $P < 0.05$, one sample *t*-test; Supplementary Fig. 4c). The changes in synaptic strength were not as large as those observed in any of the control subthreshold Up state pairing groups, but support the interpretation that synaptic weakening is under postsynaptic control. An important role for postsynaptic NMDAR in synaptic weakening following Up state pairing is consistent with many models of long-term depression (LTD)[43], although they do not appear to be necessary for presynaptic LTD induced by spike timing protocols[33–35].

Activity of glycogen synthase kinase-3β (GSK3β) can gate postsynaptic induction of NMDAR-dependent LTD[44], and, interestingly, is more active during sleep in rats[45]. The same intracellular pharmacology approach was used to assess a possible role for GSK3β in the Up state-associated synaptic plasticity (Fig. 6b). Including the GSK3β inhibitor SB415286 in the patch-pipette, at a concentration that had previously been reported to block LTD[44], also fully prevented synaptic weakening from 80.16 ± 4.1% (*n* = 6) in the vehicle control group (0.01% DMSO in internal solution) to 106.2 ± 5.2% (*n* = 6) in the inhibitor group ($t_{(10)} = 3.91$, $P < 0.01$, unpaired *t*-test). This suggests that synaptic weakening of subthreshold inputs during Up states depends on the postsynaptic activity of both NMDAR and GSK3β.

## Discussion

During slow-wave sleep, anesthesia, and even quiet wakefulness, cortical neurons display a slow synchronized membrane potential oscillation between Up and Down states, reflected as slow waves in macroscopic extracellular field potentials. How this activity pattern regulates synaptic plasticity, a likely cellular substrate for memory, has remained largely unexplored. Here, we studied the rules and mechanisms of synaptic plasticity during the Up-Down states in acute slices of mEC. We found that subthreshold inputs evoked by extracellular stimulation during Up states undergo synaptic weakening. Furthermore, we were able to demonstrate that spontaneous patterns of Up state spiking can induce synaptic weakening at identified recurrent connections between LIII pyramidal neurons. In classic spike-timing dependent plasticity, presynaptic activity that precedes postsynaptic spiking induces synaptic potentiation[46]. We found that pairing synaptic inputs with postsynaptic spiking during Up states affected synaptic strength with the same directionality: pairing inputs with single or double spikes maintained synaptic strength, while burst pairing

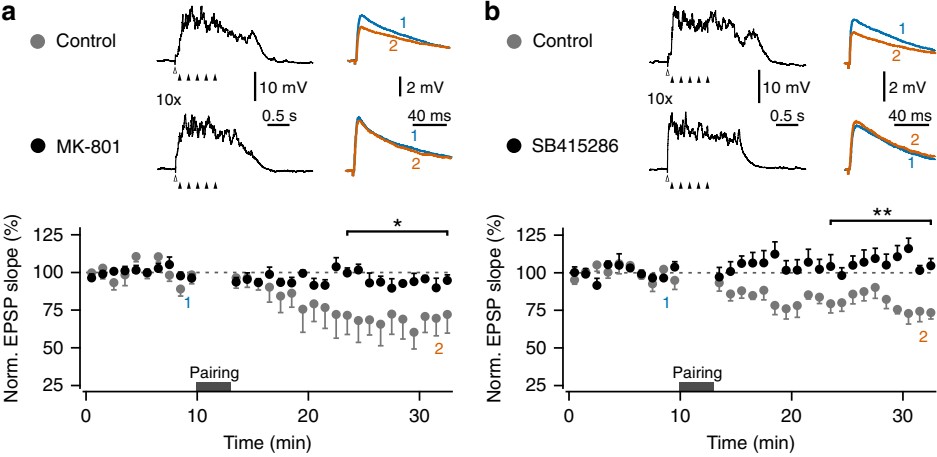

**Fig. 6** Up state-induced synaptic weakening depends on postsynaptic NMDAR and GSK3β. Single-cell pharmacology was used to explore the mechanisms of synaptic weakening induced by subthreshold Up state pairing, using extracellular stimulation protocols analogous to those in Fig. 1d (stimulus trains at 5 Hz; total of 50 individual EPSP pairings per recording). Both **a** intracellular blockade of NMDA receptors with MK-801 (200 µM in internal solution, $n = 6$ vs. vehicle control, $n = 5$) and **b** inhibition of GSK3β SB415286 (10 µM in internal solution + 0.01% DMSO, $n = 6$ vs. vehicle control, $n = 6$), significantly reduced synaptic weakening. *Top, middle*: representative recordings during pairing protocols, showing the timings of Up state induction (*white arrowheads*) and the synaptic stimuli (*black arrowheads*). *Top, right*: representative average EPSP traces from the baseline period (1; *blue*) and the end of recording (2; *orange*) (average of five traces). *Bottom*: pooled data of normalized single EPSP slope across experiments for each pairing protocol (1 min bins). **$P < 0.01$, *$P < 0.05$, unpaired $t$-test. *Error bars* show the SEM

could enable either potentiation at naïve synapse or the reversal of synaptic weakening. Interestingly, potentiated synapses appeared to be protected against subsequent Up state-induced synaptic weakening. In summary, Up states seem to promote a reduction in synaptic efficacy that can be counteracted by coordinated postsynaptic spiking. As we typically recorded for less than thirty minutes after plasticity induction, we here referred to the observed depression as 'synaptic weakening'. However, inputs typically remained weakened until the recording end at a stable level, and, thus, this synaptic weakening is likely to be a form of LTD.

The induction of synaptic plasticity is governed by multiple mutually-dependent factors, including firing rates of the pre- and postsynaptic cell, spike timing, and postsynaptic depolarization[47–49]. Even for the same synapse type, the precise patterns of activity during plasticity induction can influence the molecular pathways recruited, and result in a pre- and/or postsynaptic form of plasticity. Here, we were able to induce synaptic weakening by extracellular synaptic stimulation paired with Up states, despite predominantly low or absent postsynaptic Up state spiking. This result is in agreement with the repeated observation in neocortex that presynaptic activity coinciding with moderate postsynaptic depolarization induces LTD[50–52]. However, whereas many previous induction protocols involve an artificial modification of the membrane potential (e.g., via voltage-clamping or current injection), in our experiments, this depolarization was generated by network Up states. An independence of postsynaptic spiking for plasticity induction could be beneficial: while forms of plasticity that rely on postsynaptic spiking differentially affect synapses depending on their location[52, 53], Up states occur throughout the dendritic tree[54, 55] and, thus, properties of synaptic weakening could be more uniform within dendrites. We also showed that Up state-associated synaptic weakening induced by extracellular stimulation involves postsynaptic NMDA receptors and GSK3β. NMDAR-dependent forms of LTD are well documented, and the NMDAR-mediated $Ca^{2+}$ influx in the postsynaptic cell is a critical element in the induction process[43]. Interestingly, we find that synaptic activation during Up states leads to increased spine $Ca^{2+}$ transients compared to Down states. Moreover, GSK3β, a kinase that was previously found to be important for the induction of NMDAR-dependent LTD in

hippocampus, is also likely to be regulated by $Ca^{2+}$ signaling pathways[44, 56]. Thus, our results are consistent with a model in which NMDA receptors, and possibly other $Ca^{2+}$ sources such as voltage-gated $Ca^{2+}$ channels, mediate a boost of synaptically evoked $Ca^{2+}$ transients during Up states, which contributes to the induction of synaptic weakening[23] by activating downstream signaling pathways that involve GSK3β.

We do not have conclusive evidence as to whether synaptic weakening is pre- and/or postsynaptically expressed, but there are some indications. Analyses of changes in the CV and paired pulse facilitation, based both on our paired-recording data and on the subthreshold Up state pairings experiments with extracellular synaptic stimulation, were consistent with a presynaptic component of expression. Similar changes in the statistics of synaptic transmission have been observed for presynaptic depression induced by pairing inputs with a depolarizing current step in neocortical pyramidal cells[51]. This protocol mimics the membrane potential trajectory observed during subthreshold Up states, and we were also able to partially reproduce Up state-induced synaptic weakening in mEC with postsynaptic membrane depolarization. In order for presynaptic expression to fully explain our results, retrograde signaling to the presynaptic terminal would have to be dependent on the activation of postsynaptic NMDAR[57, 58]. However, LTD that depends on postsynaptic NMDAR is often postsynaptically expressed[43], and it has been suggested that GSK3β regulates postsynaptic AMPA receptor function and trafficking[56]. It may be that synaptic weakening induced during active network states involves a mixture of presynaptic and postsynaptic modifications. Indeed, the frequency and amplitude of miniature excitatory postsynaptic currents, measured in cortical slices of rats and mice, were found to be increased after prolonged wakefulness relative to recordings obtained after sleep[59], consistent with both presynaptic and postsynaptic changes. Further studies will be required to resolve the precise molecular mechanisms underlying the induction and expression of Up state-induced synaptic weakening, and their interaction with spike-related signaling.

Oscillations between cortical Up and Down states are a characteristic feature of slow-wave sleep. We studied state-

dependent synaptic plasticity in acute slices of mEC, which cannot mimic the complex distributed patterns of network activity that occur during natural sleep. However, our results do suggest that Up states could support elements of both synaptic homeostasis and the consolidation of synaptic memories. In our experiments, Up states promoted synaptic weakening, which could be prevented by correlated simple spiking activity. Such a plasticity mechanism is not suitable for simple synaptic scaling, but could be consistent with more selective renormalization, in which synapses are spared if they are sufficiently large or on crowded dendritic branches[60], and thus potentially more likely to drive postsynaptic spiking. We also found that pairing synaptic inputs with burst firing during Up states induced synaptic strengthening, which could support an additional process of consolidation. A potential role for NREM activity patterns in supporting synaptic strengthening is consistent with a recent study in the hippocampal CA1 – this brain region shows sharp-wave ripple activity during cortical slow-wave activity, and the study found that mimicking sharp waves was necessary to induce potentiation during the replay of spike patterns recorded in vivo[21]. It should be noted that when we examined the effects of Up state spike patterns generated spontaneously in vitro, we only observed variable degrees of synaptic weakening. Of course, even the preservation of synaptic weights within reactivated cell assemblies, on a background of more global synaptic weakening, could alone mediate a form of consolidation.

A potential link between our results and sleep-related synaptic plasticity could be provided by the involvement of GSK3β activity. This kinase has been implicated in memory reconsolidation[61], is more active during sleep[45], and would be expected to promote synaptic weakening. Moreover, GSK3β activity can be locally inhibited following LTP[44], which could explain why we were unable to induce Up state-induced weakening following input-burst induced potentiation, and offer a biochemical level of protection for recent synaptic memories. Interestingly, increased GSK3β activity has also been implicated in Alzheimer's disease (AD) pathology, including the disruption of synaptic plasticity[62, 63] and memory processing[64]. While there is likely to be a complex interaction between AD pathology, sleep, and circuit dysfunction[65], it is possible that GSK3β over-activity in AD could preclude the fine tuning of synapses during slow-wave sleep.

In order to more fully understand the implications of our results, it will be important to determine whether the features of state-dependent synaptic plasticity observed here extrapolate to other brain regions and life stages, operate in vivo during slow-wave activity, and are capable of inducing persistent changes in synaptic structure and function. Slow-wave activity has also been suggested to provide privileged periods for homeostatic regulation of intrinsic excitability[66], neuronal energy homeostasis, cellular repair, and the clearance of toxic substances and metabolites from the brain[67, 68]. To what extent each of these processes might contribute to the beneficial effects of sleep has also yet to be determined. However, this study characterizes the rules and mechanisms of synaptic plasticity that can operate during Up/Down states, and could thus provide a crucial mechanistic entry point for future research attempting to establish links between synaptic tuning and the higher-level functions of slow-wave sleep.

## Methods

**Animals and acute-slice preparations**. All procedures involving animals were carried out in accordance with the UK Animals (Scientific Procedures) Act 1986. C57BL/6 mice (P14–P21) were decapitated under deep isoflurane-induced anesthesia, and the brains placed in ice-cold artificial CSF (aCSF) containing (in mM): 126 NaCl, 3.5 KCl, 1.25 $NaH_2PO_4$, 1 $MgSO_4$, 2 $CaCl_2$, 26 $NaHCO_3$, and 10 glucose, at pH 7.2–7.4 when bubbled with carbogen gas (95% $O_2$, 5% $CO_2$). Horizontal brain slices (350 μm) were prepared using a Leica Vibratome VT1200S, and transferred to an interface chamber filled with room temperature

carbogenated aCSF to recover for at least 1 h. For recordings, slices were transfered to submerged chamber, and superfused with carbogenated aCSF heated to 32–34 °C.

**Electrophysiology**. Whole-cell patch-clamp recordings from mEC layer III principal cells were obtained with standard borosilicate glass micropipettes (6–8 MΩ containing (in mM): 110 potassium-gluconate, 40 HEPES, 2 ATP-Mg, 0.3 GTP, 4 NaCl (pH 7.2; 270–290 mOsmol $l^{-1}$). Biocytin (Sigma-Aldrich) was added at 4 mg $ml^{-1}$ to allow post hoc assessment of cell morphology. For local field potential recordings, a micropipette (4–6 MΩ) was filled with aCSF and placed in mEC layer III. In combined electrophysiology and multiphoton imaging experiments, current-clamp recordings were carried out using an Axon Multiclamp 700A amplifier (Molecular Devices) and digitized using an Axon Digidata 1440A, with data acquisition and stimulation protocols controlled using Axon pClamp. For all other experiments, recordings were obtained using an Axon Multiclamp 700B amplifier and digitized using an ITC-18 A/D board (Instrutech), with data acquisition and stimulation protocols controlled using custom-written procedures in IgorPro (WaveMetrics). Whole-cell recordings and local field potential (LFP) signals were low-pass filtered at 3 kHz, and acquired at 10 kHz. LFP signals were subsequently low-pass filtered at 300 Hz. The input resistance was determined by a hyperpolarizing 20 pA step for 300 ms in each sweep. Experiments were excluded if the input resistance or membrane potential from the end measurement period after pairing changed by more than 30% and 8 mV, respectively, compared to the baseline period. Changes in intrinsic properties did not occur systemically within each experimental group, and across the subthreshold pairing groups, the mean change input resistance was within ±4% and mean change in membrane potential was within ±2 mV (from experiments in Figs. 1d and 6)

Layer III principal cells were identified based on location, morphology, and electrophysiological characteristics. Cells displayed spiny dendrites and a thick apical dendrite extending towards layer I. The input resistance, derived from a step protocol applied at the beginning of each recording, was 207.1 ± 12.7 MΩ (n = 21; from Fig. 1d). This average input resistance measurement, the Up state amplitude (Results section), Up state duration (2.4 ± 0.2 s, n = 15; from Fig. 1d), and Up state spike rate (see Fig. 3f) are similar to previously reported in vitro and in vivo measurements from layer III principal cells in mEC[6, 28].

For paired recordings from synaptically coupled cells, both patch electrodes were placed first just over the two selected cells, followed by quick breakthroughs to avoid unnecessary cell dilution. Since the connection probability of cortical principal cells typically decreases with increasing distance between the cells, the distance between the somata of all tested pairs was chosen to be within ~150 μm. Where appropriate, current injection in the presynaptic cell was used to suppress or maintain Up state spiking. For plasticity induction, Up states were triggered at 0.067 Hz in both the presynaptic suppression group (16.20 ± 1.05 min) and presynaptic spiking Up state group (16.04 ± 1.16 min). In two recordings from each group, the sweep length did not always fully cover the Up state period. Up state spike rates from these recordings are, hence, approximations.

**Extracellular stimulation and Up state induction**. Inputs to layer III principal cells were activated with a tungsten electrode placed in the upper region of layer V (0.1 ms, 50–100 μA). Stimulation in this region gave the most consistent sub-threshold responses. We selected, if possible, for small, one-component EPSPs (during the Down state measurements). However, especially during the Up state phase, an inhibitory component often became apparent (Supplementary Fig. 1). Responses were likely evoked by stimulation of both extrinsic and intrinsic connections, including, superficially projecting layer V principal cells and antidromically activated local layer III principal cells[69].

Single Up states were evoked with a tungsten electrode placed in mEC layer III (0.1 ms, 50–200 μA pulses, typically more than 200 μm away from the recorded cell (s), as described previously[27]. These evoked Up states exhibited strikingly similar characteristics to those occurring spontaneously. Pooling subthreshold Up state pairing experiments from Fig. 1d and control recordings from Fig. 6, we manually identified 6 recordings, in which full-length spontaneous Up states were recorded during the baseline period (mean: 2.3 ± 0.6 spontaneous Up states per recording). The mean amplitude and duration of evoked Up sates was 99.3 ± 6.0% and 91.2 ± 6.5%, respectively, compared to these spontaneously occurring Up states.

For an automatic detection of Up state duration, voltage traces were median filtered (50 point window) and the membrane potential noise was calculated as the running SD over a 200 ms window. The Up state start was simply defined as the time of the induction stimulus. The Down state transition was initially detected as a decrease in membrane potential noise below 3 SD above the baseline mean, lasting for a minimum duration of 500 ms. The end of the Up state was then defined as the point at which the membrane noise had decreased to the baseline mean, or, if this fell into a phase of after-hyperpolarization, as the preceding time point at which the membrane potential reached the value of the pre-Up state membrane potential. The Up state end detected in this way was visually inspected, and manually corrected in a small minority of traces.

In Up state pairing protocols, Up states were evoked 100 ms before the first input activation. This was to ensure that the entire stimulus train would fall into Up state period, despite variations in Up state duration. All pairing protocols with extracellularly evoked synaptic activity consisted of 10 or 5 stimulus trains, with

5 or 10 stimuli per train, and with trains triggered every 15 s. Train stimulation was at 5 Hz, approximating typical Up state spike rates. In the subthreshold Up state pairings from Fig. 1d, whether the 50 total pairings were applied as 5 or 10 stimulus trains made virtually no difference on the degree of synaptic weakening achieved: $69.4 \pm 8.9\%$ ($n = 4$) and $75.6 \pm 13.1\%$ ($n = 5$), respectively, and these data were therefore combined.

Local synaptic stimulation during spine $Ca^{2+}$ imaging was achieved with a micropipette, filled with aCSF and Alexa 594 (40 µM) for placement under the fluorescence microscope. In spine $Ca^{2+}$ imaging experiments, Up states were evoked 400–500 ms before synaptic activation was triggered – this was within the train stimulation period, while ensuring that phasic $Ca^{2+}$ events could not be induced directly by the Up state stimulation electrode.

**Pharmacology**. In some patch-clamp experiments, MK-801 (200 µM; purchased from Sigma-Aldrich) and SB415286 (10 µM in internal solution + 0.01% of the vehicle DMSO; purchased from Abcam) were additionally included in the patch-pipette. The inhibitor and control internal solutions were given an initial, ~30 min, equilibration time. Half of the experiments were blinded. Up state spike rates were relatively low ($0.22 \pm 0.08$, $n = 23$). It needs to be mentioned that there are two GSK isoforms (GSK3α and GSK3β) expressed in the mammalian brain. However, since the GSK3β isoform is generally considered to be the critical one in regulating synaptic plasticity[56], we refer throughout this study to GSK3β as the main target of SB415286.

**Fluorescence imaging**. An Olympus BX51WI microscope equipped with a CAIRN Research OptoLED Lite system, optiMOS sCMOS camera (QImaging), and a 40x/0.8 NA water-objective (Olympus) was used for wide-field fluorescence imaging. A custom-built microscope[70] was used for all two-photon imaging experiments. The laser source was a pulsed Ti:Sapphire laser (Tsunami, Spectra Physics) with a peak wavelength of 850 nm. A 40×/0.8 NA water-objective (Olympus) was employed for live-cell two-photon imaging, while a 40×/1.15 NA water-objective (Olympus) was used for high-resolution imaging of fixed slices.

In spine $Ca^{2+}$ imaging experiments, cells were loaded with internal solution containing Alexa Fluor 594 (40 µM) and either the $Ca^{2+}$ indicator OGB-1 (200 µM; wide-field imaging) or Fluo-5F (400 µM; two-photon imaging) for at least 30 min. Fluorescent probes were purchased from ThermoFisher Scientific. Small EPSPs were evoked by local electrical stimulation (probably activating only a few spines). Wide-field fluorescent image series containing the target spine (20 Hz frame rate), or two-photon line-scans through the target spine, were recorded every 10–15 s. In spine $Ca^{2+}$ boost experiments investigated with wide-field microscopy, there was no indication that the target spines were spontaneously active during Up states (tested with a mean of $5.4 \pm 0.6$ successful Up state inductions without EPSP trigger per experiment), although $Ca^{2+}$ transients could occasionally be observed in other non-target spines within the field of view. During two-photon imaging, spontaneous $Ca^{2+}$ transients during Up states in the imaged spine were only seen in one recording, and were clearly separated from the evoked response. We typically recorded <20 responses in total per spine to avoid any effects of synaptic weakening and phototoxicity.

**Data analysis**. The magnitude of synaptic plasticity was quantified as the baseline-normalized average EPSP measurement of the last 10-min period of each recording condition. Very few individual measurements were ignored if they were contaminated by spontaneous activity.

The pre-stimulus Up state amplitude is defined as the difference between the average Up state membrane potential from −10 ms to each pairing stimulus and the average baseline membrane potential before each Up state induction.

In wide-field spine $Ca^{2+}$ imaging experiments, images were first registered with Fiji and the same mask was used throughout individual experiments to quantify mean $Ca^{2+}$ responses from the respective target spine and adjacent shaft over time. Wide-field $Ca^{2+}$ traces were bleach-corrected by fitting an exponential function through the baseline region and the recording end (last 0.5 s) and correcting the original trace accordingly. The spine response was quantified as the mean $\Delta F/F$ value from the 400 ms period after synaptic stimulation (for the quantification of two-photon line-scans, a 20 ms offset was introduced due to the typically better temporal resolution). To examine the effect of the slow oscillation phase on synaptically evoked $Ca^{2+}$ transients, raw data (in $\Delta F/F$) of successful synaptic transmissions (peak $Ca^{2+}$ response > 2.5 SD of baseline noise) from each group were determined and the means compared; using wide-field and two-photon microscopy, respectively, $4.0 \pm 0.5$ and $3.6 \pm 0.4$ responses in the Down state groups and $4.1 \pm 0.4$ and $3.4 \pm 0.5$ responses in the Up state groups were on average included. To investigate the dependence of spine $Ca^{2+}$ transients on NMDAR, responses recorded ~5 min after D-AP5 or aCSF (control) bath application were normalized to the respective baseline response (mean of 5 consecutive traces each).

Data were analyzed with Igor Pro (WaveMetrics) and Fiji/ImageJ, and statistics were performed with Prism (GraphPad Software) and SPPS (IBM). Applied statistical tests (one-sample, paired, and unpaired two-tailed Student's $t$-test, one-way analysis of variance, and mixed measures ANOVA) are specified in the results sections. Following one-way analysis of variance, significance was tested using a Tukey's post hoc test, and pairwise Cohen's $d$ values were calculated as the difference between group means divided by the estimate of the pooled standard deviation. Data are presented as mean ± SEM.

**Data availability**. Supporting data are available on request.

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

## Acknowledgements

This research was supported by a BBSRC grant (BB/J018074/1) to E.O.M. and T.W.; a Wellcome Trust PhD scholarship to J.B.; and a Berrow Foundation Lord Florey Scholarship and Janggen-Pöhn-Stiftung Scholarship to M.C.K.

## Author contributions

conceptualization by O.P., T.W., and E.O.M.; investigation by J.B., M.C.K., and S.T.; writing – original draft by J.B. and E.O.M.; writing – review and editing by J.B., M.C.K., O.P., and E.O.M.; visualization by J.B.; funding acquisition by T.W. and E.O.M.; supervision by T.W. and E.O.M.

## Additional information

**Competing interests:** The authors declare no competing financial interests.

