## [Peer Review File · Nature Communications]

Reviewers' Comments:

Reviewer #1 (Remarks to the Author):

In this study, Bartram et al address how slow-wave activity influences synaptic plasticity in the entorhinal cortex. There are two hypotheses: (1) SWA drives reactivation of memory traces, supporting redistribution and transformation to long-term memory, (2) SWA induces global downscaling of synaptic weights, which compensates for experience-dependent increases in synaptic weight during wake. The authors induce Up states in L3 principal neurons in acute slices from the EC of P14-21 mice using a brief, strong current injection at the deep layers. Using whole-cell recordings and synaptic stimulation, the authors show that subthreshold inputs are weakened when paired with Up states, whereas suprathreshold inputs are maintained. No change is observed when subthreshold inputs are paired with Down states. The authors show that Up state-induced synaptic weakening is associated with changes in CV and PPF, is inversely correlated with pre- and postsynaptic spike frequency, and is dependent upon signaling through postsynaptic NMDARs and GSK3beta.

Overall, the manuscript provides new information concerning the state-dependent regulation of synaptic plasticity in the EC. However, one main issue is that the relevance of the experiments to SWA is not clear. In addition, the plasticity rules that are uncovered by these studies are somewhat confusing, as presented, and the mechanistic description is limited. Several issues remain to be addressed before the manuscript is ready for publication.

Major Points

1. The title suggests that the authors have studied subthreshold synaptic inputs during cortical slow waves; however, it is not clear that the network Up states they induced are particularly relevant to cortical slow waves, as opposed to any other type of oscillation that would generate membrane potential depolarization. Furthermore, slow wave oscillations are on the order of 0.5-4 Hz. If they are trying to model slow wave activity, why not induce Up states correspondingly, instead of only once every 15 sec (or less)? Might this difference in timing influence the results? Indeed, considering that depolarization alone is sufficient to induce synaptic weakening of active inputs (Suppl Fig 4c), doesn't this argue that persistent network activity is not necessary in this form of synaptic plasticity? The title should accurately reflect the results of the paper.

2. The Up state in Fig 1d(ii) appears not to be associated with spiking (although Suppl Fig 1, does show spiking in this paradigm...), in contrast to those in Figs 1b, 2b, 2c. The authors argue that this is likely due to disynaptic inhibitory components induced by the extracellular stimulation. In that case, are the results obtained using this paradigm relevant to what would be observed in vivo? Please discuss. To clarify these issues, Suppl Fig. 1 should have a longer time scale (up to 200 delta t) and should include similar graphs for subthreshold Down state and suprathreshold Up state as well.

3. It is not clear why the authors chose to focus selectively controlling the presynaptic (and not postsynaptic) spike rate for the experiments in Fig 2 – more explanation to introduce this choice is necessary. Doing an equivalent analysis from the postsynaptic perspective could be very informative.

4. The authors see a counterintuitive decrease in synaptic weakening with increasing presynaptic spike rates. They suggest that this is due to a corresponding increase in postsynaptic spike rates, which could lead to more spike pairings that would counteract synaptic weakening. Does this result suggest that the effect of SWA on plasticity at these synapses will depend upon the strength of the learning stimulus? That is, with weaker learning stimuli, will subthreshold inputs be weakened during Up states, but with stronger learning stimuli, no input-specific weakening? What is the model?

5. The calcium imaging in Fig 3 is incomplete. First, the n is too low (only 5 spines!). Please report both spine and cell numbers. Second, the authors should include plots showing how Up states influence baseline Ca. Furthermore, they should show that the responses that they are measuring are blocked by MK801. Finally, no attempt is made to establish a relationship between Ca signaling and Up state-induced synaptic weakening.

6. The authors should discuss in further detail whether their results support either of the hypotheses presented in the introduction. Furthermore, designing a more physiologically relevant paradigm to make a direct test of these hypotheses would be desirable. For example, could the authors induce LTP at the relevant synapses and then examine whether a subsequent wave-state acts to strengthen or weaken those synapses?

Minor Points:

7. The authors should explain in the results section why 5 Hz is chosen for the evoked EPSPs.

8. Please add open arrowhead indicating Up state induction to Fig 1d(ii) and (iii), and any other Figs where it is missing.

9. The authors use the word 'occlude' both in the abstract and on p.5, when stating that "correlated pre- and postsynaptic spiking was found to 'occlude' Up state-induced synaptic weakening". However, 'occlusion', as it is most commonly used in the plasticity field, suggests that they do not see additional weakening because the synapses are already weakened. Is this what the authors wished to argue? Do they mean 'prevent' instead? Does Down state pre-post spike pairing lead to LTP? If so, that would suggest additivity rather than occlusion.
10. The authors use the word 'de-depression' to describe the results of pairing synaptic activation with spike bursts during Up states following synaptic weakening (Suppl Fig 2). However, are they simply observing potentiation? Is a similar level of potentiation achieved even in the absence of prior depression?
11. In Fig. 2B, the phrase 'presynaptic "subthreshold" Up states' is confusing, esp. because the amplitude of the Up state does not change with hyperpolarization. Perhaps "no spiking" versus "spiking" Up states would be more appropriate?
12. It is not clear why the EPSP is not visible in Fig 3a(ii). The authors should include mean dendrite $\Delta F/F$ in Fig 3c. They also should include a line on the image to show where the line scan was performed.
13. Inhibitors of GluN2B are used suggesting that observed effects are due to presynaptic NMDARs; however, no data or references are provided supporting a solely presynaptic locus for NR2B.
14. The authors should reference and discuss a new paper from the Mellor lab (Sadowski et al., 2016) that shows that reactivated place cell firing patterns induce LTP in CA1 cells only if accompanied by SWR-associated synaptic activity.
15. It would be helpful to see a list, or at least a detailed discussion, of the fundamental rules of plasticity associated with SWA that have been uncovered by this study, and what remains to be studied.

Reviewer #2 (Remarks to the Author):

This is an excellent and timely study that addresses an important unanswered question in sleep biology: what is the function of up and down states during sleep? The idea that slow waves might influence plasticity has been around for a while, but the evidence for such a function has been scant. There are some important limitations of the present study, chiefly that it relies on a

reduced preparation instead of the intact, unanesthetized brain and naturally occurring sleep. Having said that, it provides needed information about what might occur under more natural conditions.

Other than the authors acknowledging this fact more clearly in their discussion, I only have a few comments.

1. Can the authors explain more fully why they selected the stimulus train (5Hz)? Is this a pattern naturally observed during sleep in the area of cortex they investigated?
2. The authors may wish to cite Benington and Frank, 2003, as this article was one of the first to suggest that spike-timing dependent plasticity could occur during NREM sleep and also proposed a role for voltage-dependent calcium channels in sleep-mediated LTD.

Discussion

Sentence about in the middle--'predicts postsynaptic spiking' do the authors mean 'precedes'?

Reviewer #3 (Remarks to the Author):

The main goal of this study was to find how Up state affects the long term synaptic plasticity in the cortex. They found that when synaptic inputs are induced by electrical stimulation during the Up-state the response will undergo synaptic weakening but will remain unchanged if it will successfully evoke spikes. They further demonstrate that this effect is local and depends on the relative pre and post synaptic firing rates. Imaging of Ca in spines show that the network state is encoded by Ca concentration at the dendritic spines. Finally they show that this weakening depends on NMDR and GSK3 β . Taken together they suggest that slow-wave activity may bias synaptic input towards weakening, unless these inputs evoke spikes. The goals are interesting and very important and the effect of Up and Down states on synaptic plasticity, as far as I know, was never addressed directly as in this study. The experiments were designed well and the conclusions are well justified by the data. While I strongly support this study, I have a few major and minor comments that are aimed in improving the paper.

Major:

1. The protocol in Figures 1 and 4 is restricted to pairing at 5 Hz. It will be important to know how plasticity would look like with lower or higher frequencies. This is a crucial issue that should be addressed experimentally.
2. Showing that synaptic weakening is induced by spontaneous Up states is important. Spontaneous Up states can be observed in slices using suitable pharmacology and I am surprised to see that the authors did not try it or at least show that the Up states that they evoked by

electrical stimulation share major similarities with real Up states.

3. In figure 1 the authors demonstrated that a small depolarization of the cells with current injection to induce spikes canceled the effect of subthreshold pairing. It is not clear why the author did not try to repeat this experiment during the down-state (i.e, injecting current to evoke firing during the Down state).

4. The imaging data is somewhat weakly related to the other figures. It will be important to know if they observed changes in Ca signals, following pairing that leadsto weakening of synaptic response.

5. One of the authors (OP) synthesized a caged form of mk-801(tc-MK801). Please explain why uncaging of tc-MK801 was not used here as it could potentially reduce the issue of non-specific effects.

6. As far as I see, the author do not provide information about the effects of pairing on the intrinsic properties of the cells ('threshold, RP, Rin, etc.).

More Specific comments:

1. P5 Line 118: Be more specific and write "Indeed, the pairing during the SUBTHRESHOLD Up state seemed..." using 'seemed' is confusing. Either it suppressed or not. Furthermore, there is no presented data about spikes before pairing so how do we know that the suppression was due to pairing?

2. P5 l122 "and synaptic weakening following pairing did not strongly correlate with either the Up state spike rate". Not clear. Until this point it was clear that weakening is observed only when post synaptic cells did not fire at all during pairing.

3. Average synaptic responses should be presented with confidence limits (e.g., as in Figure 1C colored traces).

4. The right statistical test in Supp. Fig. 2 will be anova.

5. P8 l188 "there was a significant decrease in synaptic weakening with increasing presynaptic spike rates" the sentence is hard to read.

6. In figure 3b I am confused by the intensity plots, showing different background levels before stimulation. The first two ("i") and ("ii") show higher intensities compared to "iii)". Please explain and show a calibration bars for the color maps.

Reviewer #4 (Remarks to the Author):

The study addresses the plasticity effects of cortical slow waves, a key question in the field of sleep and memory consolidation. The paper is well written, the methods are sound and the results presented are quite novel and relevant: 1) subthreshold inputs during up states undergo synaptic weakening, 2) spontaneous patterns of up state spiking induce synaptic weakening, and 3)

pairing synaptic inputs with postsynaptic spiking during up states led to synaptic strength depression (bursts) or maintenance (1-2 spikes). These claims have potential for explaining mechanistically the memory processing that occurs during slow-wave sleep. Therefore, to the extent that the results are solid, they should be interesting to a wide community.

My main concern has to do with precisely the point of reliability, since most of the key findings come from quite small samples ($N=4-7$). A study of statistical power and a larger N are in order.

Minor concerns:

- 1) As much as I like the paper, I think that the preparation is too artificial to support the current title. It should make explicit that this is an *in vitro* study;
- 2) What happens in desynchronized states? How does that related to REM sleep?
- 3) Supplementary Fig. 2 is quite important and should be included as a regular figure.
- 4) The discussion is too long but fails to present the caveats of the paper, in particular the artificiality of the preparation.

Response to Reviewers' comments:

Reviewer #1 (Remarks to the Author):

In this study, Bartram et al address how slow-wave activity influences synaptic plasticity in the entorhinal cortex. There are two hypotheses: (1) SWA drives reactivation of memory traces, supporting redistribution and transformation to long-term memory, (2) SWA induces global downscaling of synaptic weights, which compensates for experience-dependent increases in synaptic weight during wake. The authors induce Up states in L3 principal neurons in acute slices from the EC of P14-21 mice using a brief, strong current injection at the deep layers. Using whole-cell recordings and synaptic stimulation, the authors show that subthreshold inputs are weakened when paired with Up states, whereas suprathreshold inputs are maintained. No change is observed when subthreshold inputs are paired with Down states. The authors show that Up state-induced synaptic weakening is associated with changes in CV and PPF, is inversely correlated with pre- and postsynaptic spike frequency, and is dependent upon signaling through postsynaptic NMDARs and GSK3beta.

Overall, the manuscript is provides new information concerning the state-dependent regulation of synaptic plasticity in the EC. However, one main issue is that the relevance of the experiments to SWA is not clear. In addition, the plasticity rules that are uncovered by these studies are somewhat confusing, as presented, and the mechanistic description is limited. Several issues remain to be addressed before the manuscript is ready for publication.

We thank the reviewer for their helpful comments. We have completed 3 new series of experiments, and edited the manuscript, in order to address their concerns.

Major Points

1. The title suggests that the authors have studied subthreshold synaptic inputs during cortical slow waves; however, it is not clear that the network Up states they induced are particularly relevant to cortical slow waves, as opposed to any other type of oscillation that would generate membrane potential depolarization. Furthermore, slow wave oscillations are on the order of 0.5-4 Hz. If they are trying to model slow wave activity, why not induce Up states correspondingly, instead of only once every 15 sec (or less)? Might this difference in timing influence the results? Indeed, considering that depolarization alone is sufficient to induce synaptic weakening of active inputs (Suppl Fig 4c), doesn't this argue that persistent network activity is not necessary in this form of synaptic plasticity? The title should accurately reflect the results of the paper.

Up states in the mEC *in vivo* can persist across several neocortical Up state cycles, showing an average duration of ~1.8 s, and in some cases extending to ~10 s (Hahn et al, 2012; doi: 10.1038/nn.3236). Thus the Up states we induce have a similar duration to those observed *in vivo* during slow wave activity. Given the long duration of Up states in mEC, it is not possible to induce these Up states at 0.5-4 Hz. We would argue that we have a model of mEC slow waves, but agree that this a point for discussion, and have now replaced 'slow waves' with 'Up states' in the title, and when describing the network activity we observe in the results section.

In our study, we wanted to determine how Up states modulate synaptic strength. During Up states, persistent membrane depolarization is driven by barrages of excitatory and inhibitory inputs, and it is not known how synaptic plasticity operates during these activated states (with potential sources of interaction including recurrent inhibition, conductance state and transmitter spillover). The Up state pairing paradigm was performed in a total of 31 experiments, giving a mean depression of

75.9 % (standard deviation: 20.6; 95 % confidence intervals for the mean: [68.3, 83.4]). Pairing synaptic inputs with somatic current injection induced synaptic weakening to 88.9 ± 3.6 %, and we agree that membrane depolarization is likely to be a major factor in driving synaptic weakening. Our major conclusion is that Up states are sufficient to induce reliable weakening of subthreshold synaptic inputs. The term 'Selective' in our title referred to subthreshold versus suprathreshold inputs, and we did not intend to imply that Up states are necessary for synaptic weakening – we have therefore changed the title to 'Cortical Up states promote the weakening of subthreshold synaptic inputs'.

2. The Up state in Fig 1d(ii) appears not to be associated with spiking (although Suppl Fig 1, does show spiking in this paradigm...), in contrast to those in Figs 1b, 2b, 2c. The authors argue that this is likely due to disynaptic inhibitory components induced by the extracellular stimulation. In that case, are the results obtained using this paradigm relevant to what would be observed in vivo? Please discuss. To clarify these issues, Suppl Fig. 1 should have a longer time scale (up to 200 delta t) and should include similar graphs for subthreshold Down state and suprathreshold Up state as well.

When presenting pairing during suprathreshold Up states on a coarse time scale, it is difficult to resolve the timing of spiking and evoked synaptic responses. We chose to present examples without spikes, to emphasise that the synaptic events did not induce spiking directly during pairing, and rather present the spike correlations at a higher temporal resolution. This is now highlighted in the figure legend.

The presence of disynaptic inhibition during pairing with extracellular stimulation could influence plasticity, but instantaneous excitatory-inhibitory balance is a feature of spontaneous and sensory-evoked activity in cortical neurons *in vivo* (eg Okun & Lampl, 2008, doi:10.1038/nn.2105), and we have performed paired recordings to show that weakening occurs during spontaneous spiking activity.

To clarify these issues, spike time histograms on a longer timescale (± 100 ms) have now been included in Suppl Fig 1, with an additional panel for suprathreshold Up states. The equivalent histograms for Down state pairings contained no spikes, which is described in the legend.

3. It is not clear why the authors chose to focus selectively controlling the presynaptic (and not postsynaptic) spike rate for the experiments in Fig 2 – more explanation to introduce this choice is necessary. Doing an equivalent analysis from the postsynaptic perspective could be very informative.

We agree that examining the effect of manipulating postsynaptic activity is informative, and have examined this in single spike-pairing (Fig 1) and burst spike-pairing paradigms (data in new Fig 2) with single neuron recordings. Achieving long-term stable recordings from synaptically-connected pairs of neurons is a challenging task, particularly with the low connectivity rate in mEC. We agree that testing the effects of spike-pairings again with paired recordings would be interesting, but consider it beyond the scope of the current study, which focuses on the mechanisms of synaptic weakening.

The data from Fig 1 suggest that synapses that are active in the Up state undergo weakening, if they do not precede postsynaptic spiking. A problem of interpreting the extracellular stimulation data was that we did not know the source of the fibres being stimulated, or the spontaneous patterns of activity in these fibres during Up states. By hyperpolarizing the presynaptic neuron during Up state pairing, we wanted to test directly whether inactive synapses might undergo weakening due to correlated

change in pre- and postsynaptic membrane potential, heterosynaptic interactions or a process of global downscaling. We find that synapses that are inactive during Up states are preserved, while spontaneous patterns of synaptic activity during Up states are sufficient to drive synaptic weakening. The rationale for manipulating presynaptic spike rates is now clarified in the results section (lines 193-6).

4. The authors see a counterintuitive decrease in synaptic weakening with increasing presynaptic spike rates. They suggest that this is due to a corresponding increase in postsynaptic spike rates, which could lead to more spike pairings that would counteract synaptic weakening. Does this result suggest that the effect of SWA on plasticity at these synapses will depend upon the strength of the learning stimulus? That is, with weaker learning stimuli, will subthreshold inputs be weakened during Up states, but with stronger learning stimuli, no input-specific weakening? What is the model?

Our suggestion is that Up states enable a punitive plasticity rule, whereby synapses that are reactivated during Up states undergo weakening if they fail to induce postsynaptic spiking.

We do not find a correlation between the strength of evoked EPSPs and the degree of plasticity (lines 127-130), but we agree that strong and correlated inputs would be expected to drive postsynaptic spiking most effectively, and thus be preserved.

Our new data (in response to point 6 below) suggests there may be another level biochemical protection for recently potentiated synapses, as we do not observe synaptic weakening with subthreshold Up state pairings following synaptic potentiation (new Fig 2).

5. The calcium imaging in Fig 3 is incomplete. First, the n is too low (only 5 spines!). Please report both spine and cell numbers. Second, the authors should include plots showing how Up states influence baseline Ca. Furthermore, they should show that the responses that they are measuring are blocked by MK801. Finally, no attempt is made to establish a relationship between Ca signaling and Up state-induced synaptic weakening.

We have completed a new set of Ca^{2+} imaging experiments, using wide-field epifluorescence microscopy (similar to the approach used by Popovic et al, 2015, doi:10.1038/ncomms9436), which enabled us to search more easily for responsive spines. We report new imaging data from 8 spines, confirming a boost in spine Ca^{2+} by Up states (new Fig 4). The figure now includes trials with Up states alone (new Fig 4). We further show that blocking NMDAR abolishes synaptically-evoked Ca^{2+} transients (new Fig 5).

For both the wide-field and 2-photon imaging data sets, each spine is from a different neuron, and this is now reported in the results and figure legend.

Our imaging data demonstrate that the amplitude of spine Ca^{2+} transients is state-dependent, providing a potential chemical signature for Up state-induced synaptic weakening. Following the experiments suggested by the reviewer, we can now conclude that both spine Ca^{2+} transients in mEC LIII principal cells and Up state-induced synaptic weakening share NMDAR-dependence.

6. The authors should discuss in further detail whether their results support either of the hypotheses presented in the introduction. Furthermore, designing a more physiologically relevant paradigm to make a direct test of these hypotheses would be desirable. For example, could the authors induce LTP at the relevant synapses and then examine whether a subsequent wave-state acts to strengthen or weaken those synapses?

In order to explore the interaction between LTP and Up state-induced synaptic weakening, we performed additional experiments in which we first induced synaptic potentiation by input-burst pairings during Up states, and examined the subsequent effect of subthreshold Up state pairings at these synapses. The synaptic responses were compared to a control group, that received only the conditioning stimulus of input-burst pairings during Up states. Following potentiation, we did not find a significant subthreshold pairing ($F[1,11] = 0.01$, $p = 0.96$; $\eta^2 = 0.001$), or interaction between time and pairing paradigm ($F[1,11] = 0.02$, $p = 0.90$; $\eta^2 = 0.001$; mixed ANOVA). This suggests that subthreshold Up state pairing does not reverse potentiation. These data are presented in the results and new Fig 2.

Overall, our experiments suggest that Up states can promote synaptic weakening of subthreshold inputs, while suprathreshold inputs are preserved, protected, and sometimes strengthened. We suggest that this is not consistent with a role for Up/Down states in a global down-scaling of synaptic strength during slow wave sleep. However, this data could be consistent with more selective renormalization, where spines are spared if they are sufficiently large or on crowded dendritic branches (de Vivo et al, 2017, doi: 10.1126/science.aah5982), and thus potentially more likely to drive postsynaptic spiking.

The relevance of our data to synaptic homeostasis versus synaptic consolidation hypotheses is now highlighted more clearly in the discussion (lines 455-477).

Minor Points:

7. The authors should explain in the results section why 5 Hz is chosen for the evoked EPSPs.

A pairing stimulation frequency of 5 Hz was chosen, as this approximates Up state spike rates *in vivo* (Hahn et al, 2012; doi: 10.1038/nn.3236 – mean spike rate: 5.9 ± 0.30 , $n = 41$), and this is now stated in the results. We also shown that significant synaptic weakening can be induced by Up state pairing at 2.5 Hz (72.2 ± 5.2 % of baseline EPSP slope) and 7.5 Hz (72.0 ± 6.0 % of baseline EPSP slope), to demonstrate that the choice of stimulation frequency within the typical range of Up state spike rates is not a critical parameter.

8. Please add open arrowhead indicating Up state induction to Fig 1d(ii) and (iii), and any other Figs where it is missing.

Arrowheads are now included throughout.

9. The authors use the word 'occlude' both in the abstract and on p.5, when stating that "correlated pre- and postsynaptic spiking was found to 'occlude' Up state-induced synaptic weakening". However, 'occlusion', as it is most commonly used in the plasticity field, suggests that they do not see additional weakening because the synapses are already weakened. Is this what the authors wished to argue? Do they mean 'prevent' instead? Does Down state pre-post spike pairing lead to LTP? If so, that would suggest additivity rather than occlusion.

We apologise for the confusion here. We did not wish to argue the synapses were already weakened, and have now replaced 'occlude' with 'prevent'.

We have not systematically tested the effects of Down state pre-post spike pairing, as having established the state-dependence of synaptic weakening, we focused on the rules and mechanisms by which Up states modulate synaptic strength, as these are the periods that contain the majority of pairings during Up/Down states. However, in the experimental series to address point 6, we did perform some preliminary experiments with Down state pre-post pairings – we did not find reliable potentiation, and decided to move to the Up state burst pairing paradigm that had previously worked for the experiments in Supp Fig2. It is not clear that more Down state pre-post spike pairing experiments would be able to resolve the question of additivity vs interference – if 2 separate processes of weakening and strengthening are occurring following suprathreshold Up state pairings, it could be that the Up state enhances potentiation, and thereby cancel the effect of weakening. We believe that the reviewer's suggestion that suprathreshold pairings 'prevent' synaptic weakening describes that result, and is sufficiently agnostic regarding potential additivity. Furthermore, the double pairing experiments performed to address point 6 have provided more convincing evidence to suggest that the processes of strengthening and weakening are not independent.

10. The authors use the word 'de-depression' to describe the results of pairing synaptic activation with spike bursts during Up states following synaptic weakening (Suppl Fig 2). However, are they simply observing potentiation? Is a similar level of potentiation achieved even in the absence of prior depression?

We chose to use the term 'de-depression' to be conservative, hinting that it might be a form of potentiation. This term has now been removed, and we simply state that weakening could be reversed.

We agree with the reviewer that it was important to establish whether potentiation could be induced during Up states. We now show that Up state burst spike-pairings at naïve synapses can indeed induce significant increases in synaptic strength (new Fig 2).

11. In Fig. 2B, the phrase 'presynaptic "subthreshold" Up states' is confusing, esp. because the amplitude of the Up state does not change with hyperpolarization. Perhaps "no spiking" versus "spiking" Up states would be more appropriate?

We agree that the panel labels were unclear. As there were some spikes in the hyperpolarized group, we have changed the labels to 'Up states with presynaptic spiking suppressed' and 'Up states with spontaneous spiking' (new Fig 3).

12. It is not clear why the EPSP is not visible in Fig 3a(ii). The authors should include mean dendrite $\Delta F/F$ in Fig 3c. They also should include a line on the image to show where the line scan was performed.

The EPSP evoked during the Up state is small and brief, likely due to the presence of a disynaptic inhibitory component. This is similar to response shown in Supp Fig 1. The figures showing the imaging data have been rearranged, and include mean responses for spine and dendrite, as well as the region of interest used for analysis of wide-field imaging and the line scan path used for 2-photon imaging.

13. Inhibitors of GluN2B are used suggesting that observed effects are due to presynaptic

NMDARs; however, no data or references are provided supporting a solely presynaptic locus for NR2B.

There is evidence for presynaptic GluN2B-containing NMDAR in entorhinal cortex (cited in main text and supplementary information), particularly in young rodents (Yang et al, 2006, doi: 10.1523/JNEUROSCI.4413-05.2006). We agree that we do not have the data supporting a solely presynaptic locus. However, we show that blocking GluN2B-containing NMDAR does not block synaptic weakening. This suggests that blocking a population of NMDAR that at least includes the presynaptic NMDAR, is not necessary for synaptic weakening.

14. The authors should reference and discuss a new paper from the Mellor lab (Sadowski et al., 2016) that shows that reactivated place cell firing patterns induce LTP in CA1 cells only if accompanied by SWR- associated synaptic activity.

This study is now referenced and discussed (lines 455-477).

15. It would be helpful to see a list, or at least a detailed discussion, of the fundamental rules of plasticity associated with SWA that have been uncovered by this study, and what remains to be studied.

The discussion has now been edited to address this more directly. We highlight that Up states could support elements of both synaptic homeostasis and consolidation, as inputs that consistently evoke subthreshold responses during Up states undergo synaptic weakening, while suprathreshold inputs are preserved, protected, and sometimes strengthened. We have also discussed how further studies will be required to address whether state-dependent synaptic plasticity can be extrapolated to other brain regions and life stages, operates *in vivo* during slow wave activity, is capable of inducing persistent changes in synaptic structure and function, and, ultimately, whether it contributes to the beneficial effects of sleep.

Reviewer #2 (Remarks to the Author):

This is an excellent and timely study that addresses an important unanswered question in sleep biology: what is the function of up and down states during sleep? The idea that slow waves might influence plasticity has been around for a while, but the evidence for such a function has been scant. There are some important limitations of the present study, chiefly that it relies on a reduced preparation instead of the intact, unanesthetized brain and naturally occurring sleep. Having said that, it provides needed information about what might occur under more natural conditions.

Other than the authors acknowledging this fact more clearly in their discussion, I only have a few comments.

We thank the reviewer for their positive comments on the study. We have now highlighted more clearly that this study was performed in acute slices, and discussed the limitations of this approach in the Discussion section.

1. Can the authors explain more fully why they selected the stimulus train (5Hz)? Is this a pattern naturally observed during sleep in the area of cortex they investigated?

A pairing stimulation frequency of 5 Hz was chosen, as this approximates Up state spike rates *in vivo* (Hahn et al, 2012; doi: 10.1038/nn.3236 – mean spike rate: 5.9 ± 0.30 , $n = 41$), rather than a rhythmic pattern observed in the mEC during slow wave activity. We now show that significant synaptic weakening can be induced by Up state pairing at 2.5 Hz (72.2 ± 5.2 % of baseline EPSP slope) and 7.5 Hz (72.0 ± 6.0 % of baseline EPSP slope) and is thus not specific to 5 Hz. Furthermore, the paired recording data demonstrate that synaptic weakening can be induced by spontaneous spiking activity (mean spike rate: 4.1 ± 0.6 Hz), which lacks a rhythmic structure.

2. The authors may wish to cite Bennington and Frank, 2003, as this article was one of the first to suggest that spike-timing dependent plasticity could occur during NREM sleep and also proposed a role for voltage-dependent calcium channels in sleep-mediated LTD.

This pertinent article is now cited.

Discussion

Sentence about in the middle--'predicts postsynaptic spiking' do the authors mean 'precedes'?

We have changed 'predicts' to 'precedes', as suggested.

Reviewer #3 (Remarks to the Author):

The main goal of this study was to find how Up state affects the long term synaptic plasticity in the cortex. They found that when synaptic inputs are induced by electrical stimulation during the Up-state the response will undergo synaptic weakening but will remain unchanged if it will successfully evokes spikes. They further demonstrate that this effect is local and depends on the relative pre and post synaptic firing rates. Imaging of Ca in spines show that the network state is encoded by Ca concentration at the dendritic spines. Finally they show that this weakening depends on NMDR and GSK3 β . Taken together they suggest that slow-wave activity may bias synaptic input towards weakening, unless these inputs evoke spikes. The goals are interesting and very important and the effect of Up and Down states on synaptic plasticity, as far as I know, was never addressed directly as in this study. The experiments were designed well and the conclusions are well justified by the data.

While I strongly support this study, I have a few major and minor comments that are aimed in improving the paper.

We thank the reviewer for their positive comments on the study, and suggestions for improving the paper.

Major:

1. *The protocol in Figures 1 and 4 is restricted to pairing at 5 Hz. It will be important to know how plasticity would look like with lower or higher frequencies. This is a crucial issue that should be addressed experimentally.*

We now show that significant synaptic weakening can be induced by Up state pairing at 2.5 Hz (72.2 ± 5.2 % of baseline EPSP slope) and 7.5 Hz (72.0 ± 6.0 % of baseline EPSP slope), and that this plasticity can thus operate across the range of

typical Up state spike rates observed *in vivo* (Hahn et al, 2012; doi: 10.1038/nn.3236 – mean spike rate: 5.9 ± 0.30 , $n = 41$).

2. Showing that synaptic weakening is induced by spontaneous Up states is important. Spontaneous Up states can be observed in slices using suitable pharmacology and I am surprised to see that the authors did not try it or at least show that the Up states that they evoked by electrical stimulation share major similarities with real Up states.

Up states in the mEC *in vivo* have an average duration of ~ 1.8 s, and in some cases extend up to ~ 10 s in duration (Hahn et al, 2012; doi: 10.1038/nn.3236). The spontaneous and evoked Up states we observe in mEC slices were similar to each other, and consistent with the extended Up states observed in the mEC *in vivo*. We have now directly compared the properties of spontaneous and evoked Up states in our preparation, and reported these in the methods.

We agree that it would also be interesting to examine whether synaptic weakening could be induced by spontaneous Up states. In our own preparation, spontaneous Up state activity was rare, which enabled exquisite control of baseline and pairing conditions, but precluded testing the effects of spontaneous Up states on synaptic strength. One potential scenario would be to record baseline synaptic strength, induce a defined period spontaneous Up state activity, and subsequently measure any changes in synaptic strength. As the reviewer points out, it is possible to induce Up states in neocortex using pharmacological agents, such as carbachol (Lorincz et al, 2006, doi: 10.1523/JNEUROSCI.3603-14.2015), but we are not aware of a similar established model in the mEC. Furthermore, cholinergic activation has been reported to induce synaptic plasticity, and regulate the induction of plasticity, independently of effects on network state (for example, see references in Ruivo & Mellor, 2013, doi: 10.3389/fnsyn.2013.00002). We feel that establishing a model for pharmacological modulation of Up state frequency, and dissociating the pharmacological and state-dependent effects on plasticity, would be an interesting follow-up study.

3. In figure 1 the authors demonstrated that a small depolarization of the cells with current injection to induce spikes canceled the effect of subthreshold pairing. It is not clear why the author did not try to repeat this experiment during the down-state (i.e, injecting current to evoke firing during the Down state).

Having established the state-dependence of synaptic weakening, we focused our experiments on resolving whether (i) correlated Up state spiking activity could prevent/reverse synaptic weakening, and (ii) the mechanisms of Up state-induced synaptic weakening. We focused on plasticity during Up states, as these are the periods that contain the majority of spontaneous spikes during slow wave activity.

4. The imaging data is somewhat weakly related to the other figures. It will be important to know if they observed changes in Ca signals, following pairing that leads to weakening of synaptic response.

It would certainly be interesting to perform Ca^{2+} imaging of spines during both baseline conditions, and then following pairing that leads to weakening of synaptic responses. We had not performed these experiments because: (i) calcium chelation by the dyes is likely to influence synaptic plasticity mechanisms (for example, Harney et al, 2006, doi: 10.1523/JNEUROSCI.2753-05.2006), and (ii) the aim of the Ca^{2+} imaging experiments was to explore the mechanisms of induction, rather than any changes in spine Ca^{2+} transients following weakening.

However, we have now completed a new set of Ca^{2+} imaging experiments, using wide-field epifluorescence microscopy (similar to the approach used by Popovic et al., 2015, doi:10.1038/ncomms9436). Using this approach, we have confirmed that Up states boost spine Ca^{2+} transients, and have further shown that synaptically-evoked Ca^{2+} transients depend on NMDAR (new Fig 5). Having previously shown that Up state induced synaptic weakening is activity-dependent at the monosynaptic level, and is dependent on postsynaptic NMDAR, we believe that these new imaging experiments provide a mechanistic link between the imaging and plasticity data.

5. One of the authors (OP) synthesized a caged form of mk-801(tc-MK801). Please explain why uncaging of tc-MK801 was not used here as it could potentially reduce the issue of non-specific effects.

Unfortunately, the stock of tc-MK801 is now exhausted, and there are no plans to synthesize more of this compound, which would require a significant new investment. However, we believe that the conclusions of our paper are not affected by the use of conventional MK801 in our study. Indeed, the use of a caged compound was required only to demonstrate the precise spatial location of NMDA receptors required for synaptic plasticity (Rodríguez-Moreno et al., 2011, doi: 10.1523/JNEUROSCI.0274-11.2011), as it had been reported that depolarisation via dendritic NMDAR could electronically modulate presynaptic release properties (Christie & Jahr CE, 2008, doi: 10.1016/j.neuron.2008.08.028). Our aim here is only to distinguish between presynaptic and postsynaptic NMDA receptors, and then a caged compound is not required. Thus, in our study, we show that loading the postsynaptic neuron with MK-801 blocks synaptic weakening, and we suggest that it is not critical for our conclusions to confirm that this effect is via blocking somatodendritic NMDARs, rather than the rather unlikely scenario that axonal NMDARs in the postsynaptic cell are required.

6. As far as I see, the author do not provide information about the effects of pairing on the intrinsic properties of the cells (threshold, RP, Rin, etc.).

We used hyperpolarizing current pulses to measure input resistance during the course of experiments, and the input resistance and membrane potential were used as our inclusion criteria for stable recordings (changes of less than 30 % and 8 mV, respectively; see Methods/Electrophysiology). As such, we did not feel justified in analyzing changes in these parameters. However, we did not observe systematic changes in input resistance or membrane potential following pairing, and this is now reported in the methods section.

More Specific comments:

1. P5 Line 118: Be more specific and write "Indeed, the pairing during the SUBTHRESHOLD Up state seemed..." using 'seemed' is confusing. Either it suppressed or not. Furthermore, there is no presented data about spikes before pairing so how do we know that the suppression was due to pairing?

Spike time histograms are now presented on a longer timescale (\pm 100 ms; Supp Fig 1), with an additional panel for suprathreshold Up state pairings. We have updated text and legend to clarify that background spiking activity was observed, but that each pairing stimulus itself typically evoked a subthreshold response.

2. P5 l122 “and synaptic weakening following pairing did not strongly correlate with either the Up state spike rate”. Not clear. Until this point it was clear that weakening is observed only when post synaptic cells did not fire at all during pairing.

During the Up state pairings, there was spontaneous spiking activity, but the evoked EPSPs themselves typically evoked subthreshold responses. We have now edited this section to clarify this point, and highlighted this in the figure legends.

3. Average synaptic responses should be presented with confidence limits (e.g., as in Figure 1C colored traces).

The trial-to-trial variability in EPSP responses can be considerable, as can be observed in Supp Fig 2, and is as expected. We have tried including confidence intervals for the traces in the figures, but find that it obscures the comparison of the presented EPSP traces. However, we have performed careful analysis of the variability in the EPSP measurements before and after pairing, for both the paired recording data and the subthreshold Up state pairing data (Supp Fig 3). In the results section, we have now reported the CV for the EPSP amplitudes and slopes, respectively, in order to provide the reader with a direct measurement of this the trial-to-trial variability.

4. The right statistical test in Supp. Fig. 2 will be anova.

Throughout the manuscript we normalize the EPSP measurements relative to the baseline period, and thus do not have the baseline measurements group in any of the statistical tests. For Supp Fig2, we present the data from one experiment, as in these preliminary recordings the second pairing was not delivered at the same point in each experiment. The analysis is performed on the normalized data, with only 2 time points. We agree that ANOVA is the most powerful approach, and have used in the design and analysis of the new data on input-burst pairings (new Fig 2).

5. P8 l188 “there was a significant decrease in synaptic weakening with increasing presynaptic spike rates” the sentence is hard to read.

We have replaced this sentence with ‘synaptic weakening became less prominent with higher presynaptic spike rates’.

6. In figure 3b I am confused by the intensity plots, showing different background levels before stimulation. The first two (“i”) and (“ii”) show higher intensities compared to (“iii”). Please explain and show a calibration bars for the color maps.

The intensity plots are scaled between minimum and maximum values for each plot, with the delta F/F provided in the traces below. We have now made this clear in the legend, and provided a calibration bar for the images.

Reviewer #4 (Remarks to the Author):

The study addresses the plasticity effects of cortical slow waves, a key question in the field of sleep and memory consolidation. The paper is well written, the methods are sound and

the results presented are quite novel and relevant: 1) subthreshold inputs during up states undergo synaptic weakening, 2) spontaneous patterns of up state spiking induce synaptic weakening, and 3) pairing synaptic inputs with postsynaptic spiking during up states led to synaptic strength de-depression (bursts) or maintenance (1-2 spikes). These claims have potential for explaining mechanistically the memory processing that occurs during slow-wave sleep. Therefore, to the extent that the results are solid, they should be interesting to a wide community.

My main concern has to do with precisely the point of reliability, since most of the key findings come from quite small samples (N=4-7). A study of statistical power and a larger N are in order.

We are glad that the reviewer found the study interesting, and appreciate their comments on the reliability and statistical power.

The principal finding of the paper is that pairing subthreshold synaptic inputs with Up states induces synaptic weakening. Across the different experimental groups in this study, the 5 Hz subthreshold Up state pairing paradigm was performed in a total of 31 experiments, giving a mean depression of 75.9 % (standard deviation: 20.6; 95 % confidence intervals for the mean: [68.3, 83.4]). This suggests that the principal finding of synaptic weakening is robust and reliable. We have used this population data to estimate the sample sizes required to reach a given level of power in rejecting the null hypothesis ($H_0 = 100\%$) when there is synaptic weakening ($H_1 = 75.9\%$), in experiments designed to test the effects of spiking activity or blocking molecular pathways (power calculations performed in G*Power):

(i) For one sample two-sided t-test ($\alpha=0.05$), we calculate that we would need a sample size of 6 to achieve 60 % power and a sample size of 8 to achieve 80 % power:

(iii) For a two independent sample two-sided t-test ($\alpha=0.05$), we calculate that we would need a total sample size of 18 to achieve 60 % power, and a total sample size of 26 to achieve 80 % power:

We therefore agree that some of our data sets were not sufficiently powered to enable interpretation of non-significant results. For the intracellular pharmacology experiments, we find that synaptic weakening is significantly blocked by MK801 and SB415286. For the Ro 25-6981 experiments, we wanted to explore whether the MK801 result could be explained by leakage of the drug onto presynaptic GluN2B-containing receptors – we did not find a significant effect of Ro 25-6981, and agree that this non-significant result alone is difficult to interpret. However, we can now confirm that we still have significant synaptic weakening in the presence of Ro 25-6981 ($n = 4$, $t = 4.48$, $p < 0.05$; one sample t test) and thus conclude that this drug does not prevent plasticity. We did not intend to imply that these manipulations had no effect, and the suggested power analysis has emphasised this clearly (for example, it is estimated that detecting in 50 % block of synaptic weakening in a two independent sample t -test at 80 % power would require a total sample size of 96). We have therefore revised the manuscript to ensure that statistical interpretations of non-significant results appropriately reflect the estimated power of the tests performed.

In order to test whether synaptic weakening could occur during spontaneous spiking activity, we performed paired recordings. We find a very low connectivity rate in mEC, and achieving long-term stable recordings from synaptically-connected pairs of neurons is a challenging task. We found significant synaptic weakening in paired recordings, and believe that this is sufficient to confirm the result we find repeatedly with extracellular stimulation (as highlighted above).

As the reviewer points out, an important conclusion of our study was that pairing synaptic inputs with postsynaptic spiking during Up states prevented synaptic weakening, but that the experiments to demonstrate this were underpowered. We have now performed a new series of experiments, which were designed to (i) test the effects of pairing naïve synaptic inputs with spike bursts during Up states (recorded for 20 min post burst pairing), and (ii) explore how this influenced synaptic weakening induced by subsequent subthreshold Up state pairing (relative to control group for which synaptic strength was measured continuously for >40 min post burst pairing). The statistical design was a mixed measure ANOVA (2 groups x 2 repeated measures) for the EPSP slope normalised to the baseline period. For a priori power analysis, the alternative hypothesis was that the normalised EPSP slope would be similar for both groups following burst pairing, and that subthreshold Up state pairing would induce ~25 % synaptic weakening relative to control group. Assuming ~20% standard deviation for any Up state-associated plasticity, we estimate that the within-between interaction should account for >20% of the total within-subjects variance ($\eta^2 > 0.2$; $f(U) > 0.5$). With a correlation among repeated measures of 0.5, this requires a total sample size of 12 to achieve >80% power. Total sample sizes of 12-14 would also be sufficient to provide 60-80% power for tests of residual/induced plasticity.

We aimed for a balanced total sample size >14. Many cells were lost during the required >50 min recording period, and post-hoc analysis lead to the rejection of several experiments due to changes in input resistance. In the final analysis we had 7 control experiments and 6 double pairing experiments. The results of the analysis are reported in the manuscript, but we highlight 2 points here:

(i) In the mixed measures ANOVA, we find a significant effect of time on the normalised EPSP slope ($F_{(1,11)} = 8.57$, $p = 0.014$; $\eta^2 = 0.48$), but no significant interaction between time and pairing group ($F_{(1,11)} = 0.02$, $p = 0.90$; $\eta^2 = 0.001$). We simply conclude that subthreshold pairing following prior burst pairing does not significantly affect the time course of changes in synaptic strength or reverse potentiation. The eta squared values are reported in the manuscript, in order to help the readers to interpret the null result for the interaction.

(ii) We find that pairing synaptic inputs with spike bursts during Up states leads to an increase in synaptic strength. This potentiation decays monotonically over the >40 min post pairing, but is significantly enhanced at 10-20 min ($n = 11$, $t = 5.47$, $p < 0.001$) and 30-40 min ($n = 11$, $t = 2.62$, $p = 0.044$; corrected for multiple comparisons) relative to baseline. We therefore now feel confident in concluding that pairing synaptic inputs with postsynaptic spiking during Up states prevents synaptic weakening.

We have also performed whole new set of imaging experiments to demonstrate that spine calcium changes during Up states.

In order to ensure the reader can interpret the effect sizes for each experimental manipulation, we have now provided eta squared for each ANOVA, Cohen's d for the pairwise comparisons for One-way ANOVA followed by Tukey test, and t values for each t test.

Minor concerns:

1) *As much as I like the paper, I think that the preparation is too artificial to support the current title. It should make explicit that this is an in vitro study;*

To address this concern, we have changed the title to 'Cortical Up states promote the weakening of subthreshold synaptic inputs', and highlighted in the abstract that the experiments were performed in acute slices.

2) *What happens in desynchronized states? How does that related to REM sleep?*

We agree that this is an important question, and would anticipate that changes in neuromodulatory tone during desynchronized states would alter the plasticity rules (for example, see references in Ruivo & Mellor, 2013, doi: 10.3389/fnsyn.2013.00002). The current study does not support any speculations on this, but we believe this will be an interesting avenue for future studies.

3) *Supplementary Fig. 2 is quite important and should be included as a regular figure.*

The new experiments examining the effects of input-burst pairings are now included as a main figure, which shows more convincingly that input-burst pairing during Up states can induce synaptic strengthening.

4) *The discussion is too long but fails to present the caveats of the paper, in particular the artificiality of the preparation.*

Several sections have now been removed from the discussion, and the limitations of the study have been highlighted more clearly.

Reviewer Comments:

Reviewer #1 (Remarks to the Author):

The authors have addressed my concerns.

Reviewer #2 (Remarks to the Author):

My concerns have been addressed.

Reviewer #3 (Remarks to the Author):

The paper was greatly improved. The presentation is much better and clearer. The authors addressed most of my comments and therefore I support it as is.

Reviewer #4 (Remarks to the Author):

I am satisfied with the revised manuscript.

REVIEWERS' COMMENTS:

Reviewer #1 (Remarks to the Author):
The authors have addressed my concerns.

Reviewer #2 (Remarks to the Author):
My concerns have been addressed.

Reviewer #3 (Remarks to the Author):
The paper was greatly improved. The presentation is much better and clearer. The authors addressed most of my comments and therefore I support it as is.

Reviewer #4 (Remarks to the Author):
I am satisfied with the revised manuscript.

RESPONSE TO REVIEWERS' COMMENTS:

We would like to thank the reviewers for their help in improving the manuscript. There are no additional concerns of the reviewers' to address at this stage.